# Generative Inverse Design with Abstention via Diagonal Flow Matching

**Miguel de Campos** [1]   **Werner Krebs** [2]   **Hanno Gottschalk** [1]

## Abstract

Inverse design aims to find design parameters $x$ achieving target performance $y^*$. Generative approaches learn bidirectional mappings between designs and labels, enabling diverse solution sampling. However, standard conditional flow matching (CFM), when adapted to inverse problems by pairing labels with design parameters, exhibits strong sensitivity to their arbitrary ordering and scaling, leading to unstable training. We introduce Diagonal Flow Matching (Diag–CFM), which resolves this through a zero-anchoring strategy that pairs design coordinates with noise and labels with zero, making the learning problem provably invariant to coordinate permutations. This yields substantially lower round-trip error than CFM and invertible neural network baselines across design dimensions up to $P{=}784$, including order-of-magnitude gains on several benchmarks. We develop two architecture-intrinsic uncertainty metrics, Zero-Deviation and Self-Consistency, that enable three practical capabilities: selecting the best candidate among multiple generations, abstaining from unreliable predictions, and detecting out-of-distribution targets; consistently outperforming ensemble and general-purpose alternatives across all tasks. We validate on airfoil, gas turbine combustor, scalable analytical benchmarks, a photonics inverse-design task, and an image-statistics benchmark.

## 1. Introduction

In many engineering domains, the design process involves finding geometric or parametric configurations that achieve desired performance characteristics. Traditional approaches to this problem rely on forward design, where engineers iteratively propose designs and evaluate their performance through simulations or experiments. This process can be time-consuming and may not efficiently explore the design space, and often ends with a single feasible design alternative. Inverse design reverses this paradigm: given target performance specifications, the goal is to directly generate designs that satisfy those requirements. This approach has gained significant attention in fields ranging from aerodynamics (Sekar et al., 2019) to photonics (Molesky et al., 2018) and materials science (Noh et al., 2020), where the ability to efficiently navigate complex, high-dimensional design spaces is crucial.

Recent advances in generative modeling have enabled a new paradigm: generative design, where deep learning models learn from data to generate design parameters (Ardizzone et al., 2019). These models, including variational autoencoders (VAEs), generative adversarial networks (GANs) (Chen et al., 2020), and more recently diffusion models (Ni et al., 2023), can generate novel designs conditioned on desired performance criteria. Unlike optimization-based inverse design methods that solve for a single design per query, generative models can produce diverse design candidates that all satisfy the given specifications, offering engineers multiple options to choose from based on additional constraints or preferences not captured in the performance metrics.

In this work, we develop an invertible conditional flow matching architecture (Lipman et al., 2023) for generative design that operates bidirectionally, and is capable of both generating parametric designs given performance labels and computing performance predictions from designs. However, this generative capability introduces a critical challenge: the network will produce a design for any input performance specification, even those that are physically unfeasible or lie outside the training distribution. For instance, in aerodynamic design, the model might generate an airfoil shape for lift and drag coefficients that violate fundamental physical constraints. Without a mechanism to assess the reliability of generated designs, practitioners cannot distinguish between trustworthy predictions and spurious outputs.

We address this challenge through uncertainty quantification (UQ) (Abdar et al., 2021), introducing methods to quantify the model's confidence that a generated design genuinely achieves the specified performance. We develop

---

[1]Institute of Mathematics, TU Berlin [2]Siemens Energy. Correspondence to: Miguel de Campos <campos@math.tu-berlin.de>.

*Proceedings of the 43rd International Conference on Machine Learning*, Seoul, South Korea. PMLR 306, 2026.

two architecture-specific metrics—Zero-Deviation and Self-Consistency—that exploit the bidirectional structure of our flow, and compare against general-purpose baselines including ensemble methods (Dietterich, 2000). These UQ metrics enable users to filter out unreliable designs and focus on candidates that the model confidently predicts will meet the desired specifications. We evaluate our approach on five tasks: airfoil aerodynamics (Kanchi et al., 2025), gas turbine combustor design (Krueger et al., 2024), a scalable multi-objective benchmark (Deb et al., 2005), thin film optical coating, and a high-dimensional FashionMNIST image-statistics benchmark (Xiao et al., 2017).

### 1.1. Contributions

- **Diagonal Flow Matching (Diag–CFM)**: a zero-anchoring strategy for conditional flow matching that induces a learning problem whose difficulty is provably independent of coordinate ordering. This eliminates the sensitivity to design and label permutations present in standard CFM, yielding large improvements in round-trip accuracy across design dimensions up to $P=784$.
- **Architecture-specific uncertainty metrics**: two novel methods—Zero-Deviation and Self-Consistency—that exploit Diag–CFM's bidirectional structure. Zero-Deviation is computed as a byproduct of generation with no additional forward passes. We compare against ensemble variance and flow matching loss baselines.
- **Comprehensive evaluation** on five datasets (gas turbine combustor, airfoil aerodynamics, DTLZ benchmark, thin film optical coating, and FashionMNIST image statistics), together with UQ experiments demonstrating three practical capabilities: selecting the best candidate among multiple generations, abstaining from unreliable predictions, and detecting out-of-distribution targets.

**Code availability.** A compact reference implementation of Diag–CFM with runnable DTLZ2, thin-film, and FashionMNIST examples is available at github.com/migueldecampos/diagcfm.

This paper is structured as follows: In Section 2 we discuss related work, and make our problem set-up explicit in Section 3. Section 4 introduces diagonal flow matching (Diag–CFM) and our Uncertainty Quantification (UQ) metrics. Section 5 explains experiments and results. We conclude in Section 6 with summary and outlook.

## 2. Related Work

### 2.1. Inverse Design

Inverse design refers to the class of methods that, given target performance specifications $y^\star$, aim to directly synthe-

size designs $x$ such that the forward response $f(x)$ matches $y^\star$ as closely as possible. In its most common formulation, one seeks a solution of the constrained optimization problem $\arg\min_{x\in\mathcal{X}} \mathcal{L}\big(f(x), y^\star\big) + R(x)$, where $f$ denotes a (possibly expensive) simulator or surrogate, $\mathcal{L}$ measures the mismatch between achieved and desired performance, and $R$ encodes regularization or additional engineering constraints. This framework has been widely adopted across engineering disciplines, including aerodynamic shape optimization (Sekar et al., 2019), nanophotonics (Molesky et al., 2018), and materials design (Noh et al., 2020). In many of these applications, gradients of $f$ with respect to $x$ are obtained via adjoint methods or automatic differentiation, enabling efficient gradient-based optimization even in high-dimensional design spaces. However, the inverse design problem is typically ill-posed: the mapping $f$ is many-to-one, so there may exist a manifold of designs $\{x : f(x) \approx y^\star\}$, while non-convexity leads to local minima and optimization trajectories that are sensitive to initialization and hyperparameters. In many cases, forward surrogate models like neural networks or Gaussian processes are used for inverse design tasks, see e.g. (Forrester et al., 2008; Martins & Ning, 2022). Here first a fast surrogate model is trained on expensive simulation data and the costly inverse search is then performed on the surrogate model. Despite considerable speedup, this method still suffers from being based on a forward model and in general is not capable to provide a comprehensive set of design alternatives (Krueger et al., 2024).

### 2.2. Generative Inverse Design

Recent approaches replace iterative optimization with learned mappings. Physics-informed neural networks (Raissi et al., 2019; Cai et al., 2021) and operator learning models (Lu et al., 2021; Li et al., 2021; Drygala et al., 2022; Ross et al., 2025) learn fast forward surrogates but remain unidirectional. Domain-specific generative models have been developed for applications including airfoil parameterization via GANs (Chen et al., 2020), protein structure generation via diffusion (Ni et al., 2023), and physics foundation models (Nguyen et al., 2025; Simonds, 2025), though these typically require task-specific training or prompting. Some recent works integrate physical constraints directly into the generation process by coupling generative models with optimization or iterative sampling (Giannone et al., 2023; Wu et al., 2024). For example, DFlow-SUR (Yang et al., 2025) propagates physical gradients through the flow matching ODE, actively steering generation toward feasible targets. While highly accurate, these methods reintroduce the computational cost of iterative optimization at inference time. Simulation-based inference (SBI) provides a closely related perspective: it learns amortized posterior samplers for unknown

parameters given simulator observations, and recent work applies flow matching to this setting (Wildberger et al., 2023). These methods typically use the observation or target as a conditioning variable and transport latent noise to samples from the conditional posterior. In contrast, our work follows the fully invertible paradigm (Ardizzone et al., 2019; Dinh et al., 2017; Chan et al., 2023; Chen et al., 2018), which amortizes the cost of inverse design by learning a direct, bidirectional map (Ardizzone et al., 2019; Krueger et al., 2024). We build on Conditional Flow Matching (CFM) (Lipman et al., 2023) but identify that its standard formulation is unstable for inverse problems. We resolve this via Diagonal Flow Matching (see Section 4), enabling accurate, single-pass generation without test-time optimization. We use coupling-based INNs (Dinh et al., 2017) as a further baseline in our experiments.

### 2.3. Uncertainty Quantification

Uncertainty Quantification (UQ) is a widely studied topic in machine learning research (Hüllermeier & Waegeman, 2021). Numerous UQ methods are available, e.g. MC-dropout (Gal & Ghahramani, 2016), ensemble methods (Lakshminarayanan et al., 2017). In the context of generative learning, hallucination detection (Ji et al., 2023) is a prominent topic. A few works apply this to generative learning in physics (Rathkopf, 2025; Brown et al., 2024). These however are not compatible with the approach we pursue here, as the physics based checks to detect hallucination require a full synthesis of physical states. Here we work with much smaller and faster models that are only trained on the design parameter to label relation.

## 3. Problem Setup and Preliminaries

**Design and label spaces.** We consider *parametric design problems* in which a design is represented by a vector $x \in \mathbb{R}^P$ (e.g., geometric parameters or operating setpoints), and its performance is summarized by a vector of labels $y \in \mathbb{R}^L$ with $L < P$. The strict inequality reflects the typical many-to-one nature of forward evaluation: for a given performance specification $y$, the pre-image $\{x : f(x) = y\}$ under the evaluation function $f : \mathbb{R}^P \to \mathbb{R}^L$ often contains more than one element.

We assume access to pairs $(x_i, y_i)$ drawn from a data-generating process $y = f(x) + \varepsilon$, where $f : \mathbb{R}^P \to \mathbb{R}^L$ is a (possibly expensive) evaluator based on e.g. finite element analysis, computational fluid dynamics, or some other type of high-fidelity simulator and $\varepsilon$ models measurement/simulation noise.

**Feasible targets and success sets.** Not all label vectors are physically attainable. We denote the (unknown) feasible set $\mathcal{Y}_{\text{feas}} := \{y \in \mathbb{R}^L : \exists x \in \mathbb{R}^P \text{ with } f(x) = y\}$. Given a tolerance $\tau \geq 0$ and a target $y^\star \in \mathcal{Y}_{\text{feas}}$, the *success set* in design space is $\mathcal{S}(y^\star, \tau) := \{x \in \mathbb{R}^P : \|f(x) - y^\star\| \leq \tau\}$.

**Bidirectional Modeling Objectives.** We aim to develop a model with two complementary capabilities:

1. Generative (Inverse) Direction: Given a target performance $y^\star$, sample designs $x \sim q_\theta(x \mid y^\star)$ that concentrate on $\mathcal{S}(y^\star, \tau)$, revealing the diversity of feasible solutions.
2. Predictive (Forward) Direction: Given a design $x$, predict its performance $y$. This serves both as a fast surrogate for expensive simulations and as a consistency check for generated designs.

**Dimensionality and invertible parameterization.** Since $L < P$, a smooth bijection between $x$ and $y$ cannot exist without additional latent degrees of freedom. We therefore introduce a latent variable $z \in \mathbb{R}^{P-L}$ drawn from a simple base distribution $p_B(z)$ (e.g. uniform or normal distribution), and aim to learn an invertible map $T_\theta : (z, y) \longleftrightarrow x$, so that sampling $z \sim p(z)$ and applying $x = T_\theta(z, y^\star)$ yields $x \sim q_\theta(x \mid y^\star)$, while the reverse map recovers $(\tilde{z}, \hat{y}_\theta(x)) = T_\theta^{-1}(x)$. This augmented formulation reconciles the dimension mismatch $(\dim(z, y) = P)$ and makes synthesis and analysis two directions of the same learned transformation. While $P - L$ latent dimensions suffice in principle, we show in Section 4 that augmenting to $P$ dimensions enables a simpler, more stable formulation.

**Uncertainty in Generative Design.** In probabilistic modeling, uncertainty is typically decomposed into epistemic and aleatoric uncertainty (Hüllermeier & Waegeman, 2021). For the problem of identifying unfeasible or out-of-distribution performance queries, epistemic uncertainty is of primary interest. When a user requests a design with performance $y^\star$ that is unattainable (e.g., an airfoil with impossibly high lift-to-drag ratio), we expect the model to exhibit high epistemic uncertainty. Our uncertainty quantification methods aim to capture this epistemic uncertainty, enabling practitioners to assess the trustworthiness of generated designs online, i.e. without setting up physics based simulations.

## 4. Methodology

We instantiate the bidirectional generative-predictive model using a time-dependent continuous normalizing flow trained with *conditional flow matching* (CFM). The core idea is to learn a vector field whose induced ODE flow transports the source distribution over *augmented inputs* $(z, y)$ to the empirical distribution of designs $x$, thereby yielding an invertible map $T_\theta : (z, y) \longleftrightarrow x$, so that (i) sampling $z \sim p_B$ and integrating forward produces diverse designs conditioned

on $y$, and (ii) integrating the dynamics backward from a design $x$ reconstructs $(\tilde{z}, \hat{y}_\theta(x))$ for fast, self-consistent performance prediction. We first detail the CFM objective and the induced algorithms for synthesis and analysis, as well as our new method, Diag–CFM; and then introduce our uncertainty estimators built atop this flow.

## 4.1. Conditional Flow Matching

**Setup and probability path.** Let $p_{\text{data}}(x, y)$ denote the empirical joint distribution over designs and labels, $p_B(z)$ a simple base (e.g., uniform or standard normal) on $\mathbb{R}^{P-L}$, and define the *source* distribution on $\mathbb{R}^P$ by

$$s_0 = \begin{bmatrix} z; y \end{bmatrix}, \qquad (x, y) \sim p_{\text{data}}, \ z \sim p_B.$$

The *target* sample is $s_1 = x \in \mathbb{R}^P$. We couple $(s_0, s_1)$ via the empirical pairing of $(x, y)$ with an independent $z$. For $t \in [0, 1]$, we choose a simple linear probability path $s_t = (1 - \sigma(t)) s_0 + \sigma(t) s_1$, with $\sigma(t) = t$, whose *conditional* velocity along a paired path is

$$u_t(s_t \mid s_0, s_1) = \tfrac{d}{dt} s_t = \dot{\sigma}(t)(s_1 - s_0) = (s_1 - s_0).$$

**Learning a transport vector field.** We parameterize a time-dependent vector field $v_\theta : [0, 1] \times \mathbb{R}^P \to \mathbb{R}^P$ and train it by regressing to the target conditional velocity,

$$u := (s_1 - s_0) = \begin{bmatrix} [x]_{1:P-L} - z \\ [x]_{P-L+1:P} - y \end{bmatrix} \in \mathbb{R}^P, \qquad (1)$$

along randomly sampled paths:

$$\mathcal{L}_{\text{CFM}}(\theta) = \mathbb{E}_{\substack{(x,y) \sim p_{\text{data}} \\ z \sim p_B, \ t \sim \mathcal{U}[0,1]}} \left[ \left\| v_\theta(t, s_t) - (s_1 - s_0) \right\|_2^2 \right]. \quad (2)$$

It is known that the minimizer of (2) recovers the *conditional expectation* of the target velocity $u_t$ given $(t, s_t)$; integrating the ODE $\dot{s} = v_\theta(t, s)$ from $t{=}0$ to $1$ then pushes forward the source distribution to the target distribution along the specified probability path (Lipman et al., 2023). Because ODE flows are invertible, we obtain a bidirectional model through reverse-time integration.

**Bidirectional use.** Given a trained $v_\theta$, we realize both directions as numerical integrations:

**Synthesis (inverse)** :

    Input $y^\star$, $z \sim p_B$, $s(0) = [z; y^\star]$,

    Integrate $\dot{s} = v_\theta(t, s)$ to $t{=}1$; $x \leftarrow s(1)$.

**Analysis (forward)** :

    Input $x$, $s(1) = x$,

    Integrate $\dot{s} = -v_\theta(t, s)$ to $t{=}0$; $(\tilde{z}, \hat{y}_\theta(x)) \leftarrow s(0)$.

Thus a *single* invertible flow provides conditional generation $x \sim q_\theta(x \mid y^\star)$ and a fast predictor $\hat{y}_\theta(x)$ consistent with the same transport.

**Diagonal Conditional Flow Matching** We can observe in equation (2) that the conditional flow matching loss is *component-wise*: each dimension of $v_\theta(t, s_t)$ is independently regressed to the corresponding dimension of the target velocity $(s_1 - s_0)$. Unlike typical generative modeling scenarios where one flows from pure noise to data (e.g., image generation), our augmented space construction flows from $s_0 = [z; y] \in \mathbb{R}^P$ to $s_1 = x \in \mathbb{R}^P$, that is, the last $L$ coordinates compare design components to labels $(x_{P-L+i} - y_i)$, while the first $P{-}L$ compare design components to noise $(x_i - z_i)$. Because of this, we observed a strong dependence on the arbitrary ordering of design and label coordinates: permutations that *align* numerically similar coordinates (e.g., comparable scales) reduce the loss and train more stably (see Figure 2).

This sensitivity to the ordering of the design and label vectors is an undesirable artifact, increasing the complexity of training, and negatively impacting performance. To remove this sensitivity and significantly increase performance, we *anchor* labels against zero and pair all design coordinates with latent noise. Let $R \in \{0, 1\}^{P \times (P-L)}$ be a fixed replication/assignment matrix that expands the latent to $z^\uparrow = Rz \in \mathbb{R}^P$ by repeating entries of $z \in \mathbb{R}^{P-L}$. We augment the state by $L$ extra coordinates and define

$$s_0^{\widetilde{\text{diag}}} = \begin{bmatrix} z^\uparrow; y \end{bmatrix} \in \mathbb{R}^{P+L}, \qquad s_1^{\widetilde{\text{diag}}} = \begin{bmatrix} x; 0_L \end{bmatrix} \in \mathbb{R}^{P+L}.$$

With the linear path $s_t^{\widetilde{\text{diag}}} = (1 - t) s_0^{\widetilde{\text{diag}}} + t s_1^{\widetilde{\text{diag}}}$, the target velocity is

$$u^{\widetilde{\text{diag}}} := s_1^{\widetilde{\text{diag}}} - s_0^{\widetilde{\text{diag}}} = \begin{bmatrix} x - Rz \\ -y \end{bmatrix} \in \mathbb{R}^{P+L}, \quad (3)$$

so the per-coordinate regression always matches *design* coordinates to *noise* and *label* coordinates to *zero*.

At inference, synthesis starts from $s(0) = [Rz; y^\star]$ and returns the first $P$ coordinates at $t{=}1$ as the design $x$; analysis starts from $s(1) = [x; 0]$ and reads the last $L$ coordinates at $t{=}0$ as $\hat{y}_\theta(x)$. This "zero-anchoring" eliminates the dependence on coordinate ordering (labels are never directly subtracted from design parameters), reduces scale/units mismatch in the loss, and preserves the total number of independent degrees of freedom ($\dim z = P{-}L$) because $R$ has rank $P{-}L$. However, since generating random entries is not expensive, we can simplify the setup and dispense with the need for a matrix $R$ by working with noise $z \in \mathbb{R}^P$ from a $P$-dimensional noise distribution $p_B(z)$. This yields

$$s_0^{\text{diag}} = \begin{bmatrix} z; y \end{bmatrix} \in \mathbb{R}^{P+L}, \qquad s_1^{\text{diag}} = \begin{bmatrix} x; 0_L \end{bmatrix} \in \mathbb{R}^{P+L},$$

with target velocity

$$u^{\text{diag}} := s_1^{\text{diag}} - s_0^{\text{diag}} = \begin{bmatrix} x - z \\ -y \end{bmatrix} \in \mathbb{R}^{P+L}. \quad (4)$$

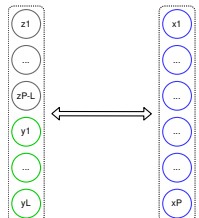
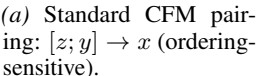
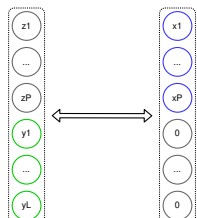

*(a)* Standard CFM pairing: $[z; y] \rightarrow x$ (ordering-sensitive).

*(b)* Diag–CFM pairing: $[z; y] \rightarrow [x; 0]$ (ordering-robust).

*Figure 1.* Per-coordinate targets in the flow-matching loss. Diag–CFM anchors labels to zero and pairs designs with latent noise, removing spurious dependence on coordinate ordering and scale.

We refer to this variant as *Diagonal* or *Zero-Anchored* (Diag–CFM). The Diag–CFM loss is

$$\mathcal{L}_{\text{Diag–CFM}}(\theta) = \mathbb{E}_{(x,y),z,t} \left\| v_\theta\big(t, s_t^{\text{diag}}\big) - u^{\text{diag}} \right\|_2^2. \quad (5)$$

**Proposition 4.1.** *The distribution of the Diag–CFM target velocity* (4),

$$U^{\text{diag}} = \begin{bmatrix} X - Z \\ -Y \end{bmatrix}_{X,Y \sim p_{data}(x,y), Z \sim p_B(z)},$$

*is equivariant with respect to permutations of label and parameter coordinates.*

**Proposition 4.2.** *The distribution of the CFM target velocity* (1),

$$U = \begin{bmatrix} [X]_{1:P-L} - Z \\ [X]_{P-L+1:P} - Y \end{bmatrix}_{X,Y \sim p_{data}(x,y), Z \sim p_B(z)},$$

*is not equivariant with respect to permutations of label and parameter coordinates.*

Proofs are provided in Appendix A. Proposition 4.1 establishes that the target velocity field is equivariant w.r.t. permutations of label and parameter coordinates. This does not imply that the model used or specific model initializations are invariant to coordinates permutations, but rather that the Diag–CFM *learning problem* is invariant to coordinate permutations. That is, that the class of functions required to approximate the target velocity is the same across all orderings, thereby decoupling the task complexity from the arbitrary arrangement of design and label vectors; a stability we validate in Figure 2 and Appendix B.

### 4.2. Uncertainty Quantification

Our generator produces designs for any requested specification $y^\star$, including unfeasible or out-of-distribution targets. We therefore require a metric indicating how confident the model is that a produced design $x$ will actually satisfy the requested performance.

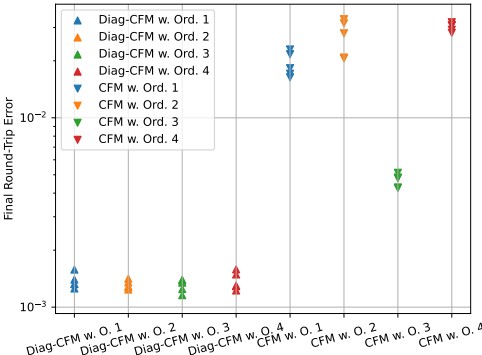

*Figure 2.* Final round-trip error (log scale) for training runs on the gas turbine dataset with different parameter orderings, comparing Diag–CFM and standard CFM (5 runs per case). CFM performance varies significantly with ordering while Diag–CFM remains stable. The gas turbine dataset is already scale-normalized to $[0, 1]$, so the observed sensitivity is attributable purely to coordinate ordering; for datasets with heterogeneous scales, the effect would likely be more pronounced. Numerical values in Appendix B.

**Ensemble Methods** Deep ensembles approximate epistemic uncertainty by training multiple models with different initializations (and optionally data resampling) and measuring the dispersion of their predictions. In our setting, let $\{\theta_m\}_{m=1}^M$ denote an ensemble of Diag–CFM models trained independently; we use $M=5$ in all experiments. We select one member, $\theta_{\text{ref}}$, as the *reference* generator.

*Generation.* Given a target $y^\star$ and a latent $z \sim p_B$, the reference model synthesizes a candidate design $x_{\text{ref}} = T_{\theta_{\text{ref}}}(z, y^\star)$.

*Evaluation by the ensemble.* Each ensemble member evaluates the same design using its analysis (forward) direction $\hat{y}^{(m)} = \hat{y}_{\theta_m}(x_{\text{ref}}) \in \mathbb{R}^L$ for $m = 1, \dots, M$. Let $\bar{y} = \frac{1}{M} \sum_m \hat{y}^{(m)}$ be the ensemble mean. We compute the sample covariance $\widehat{\Sigma}_{\text{ens}}(x_{\text{ref}}) = \frac{1}{M-1} \sum_{m=1}^M (\hat{y}^{(m)} - \bar{y})(\hat{y}^{(m)} - \bar{y})^\top$ and define a scalar *uncertainty score* as the total variance in a standardized label space: $u_{\text{ens}}(x_{\text{ref}}) = \text{tr}\left(D_y^{-1} \widehat{\Sigma}_{\text{ens}}(x_{\text{ref}})\right)$, where $D_y = \text{diag}(\sigma_1^2, \dots, \sigma_L^2)$ contains per-label variances estimated on the training set (or set $D_y = I$ if labels are pre-standardized).

**Flow Matching Loss.** Following the intuition that out-of-distribution samples exhibit larger deviations from learned dynamics (Mahmood et al., 2020), we include the flow matching loss as a baseline uncertainty metric, measuring how straight a trajectory the model's velocity field produces for a given generated sample. We evaluate the Diag–CFM loss (5) at a fixed interpolation $t=0.5$: $u_{\text{FM}}(x_{\text{ref}}) = \|v_\theta(0.5, s_{0.5}^{\text{diag}}) - u^{\text{diag}}\|_2^2$. Higher values indicate the model is poorly calibrated for this sample.

**Architecture-Specific Methods** The Diag–CFM architecture enables two additional uncertainty metrics that exploit its zero-anchoring structure, requiring only a single trained model and no ensemble.

*Zero-Deviation.* Recall that in Diag–CFM, synthesis integrates from $s(0) = [z; y^\star]$ to $s(1)$, where the target state is $[x; 0_L]$. A well-trained model should produce outputs whose last $L$ components are near zero. We define the *zero-deviation* uncertainty as $u_{\mathrm{zero}}(x_{\mathrm{ref}}) = \left\| [s(1)]_{P+1:P+L} \right\|_2^2$, where $[s(1)]_{P+1:P+L}$ denotes the last $L$ components of the synthesis output. Larger deviations indicate that the flow struggled to map the request $(y^\star, z)$ onto the learned data manifold, signaling epistemic uncertainty. Crucially, this metric is computed as a *byproduct* of generation with no additional forward passes.

*Self-Consistency.* A reliable generated design should, when passed back through the analysis direction, reconstruct the original target labels. Since the flow is an ODE, it is invertible up to numerical precision—naïvely running analysis on the synthesis output would yield a trivial round-trip. The key is that we *discard* the approximate zeros produced by synthesis and replace them with exact zeros: given $x_{\mathrm{ref}}$ from synthesis output $[x_{\mathrm{ref}}; \tilde{0}]$, we form the analysis input $s(1) = [x_{\mathrm{ref}}; 0_L]$ and integrate backward to $t{=}0$, obtaining $s(0) = [\tilde{z}; \hat{y}_{\mathrm{rec}}]$. The *self-consistency* uncertainty is $u_{\mathrm{sc}}(x_{\mathrm{ref}}) = \left\| \hat{y}_{\mathrm{rec}} - y^\star \right\|_2^2$. This metric tests whether the generated design is internally consistent with the model's learned forward mapping: a design that truly achieves $y^\star$ should reconstruct it when analyzed. Large reconstruction errors indicate the model is uncertain about the validity of the generated solution. This requires one additional forward pass through the model.

Both metrics leverage Diag–CFM's zero-anchoring architecture: zero-deviation probes the quality of the synthesis endpoint, while self-consistency probes round-trip coherence. Unlike ensemble methods, they operate on a single model. And unlike the flow matching loss zero-deviation requires no extra model evaluations.

# 5. Experiments

We evaluate Diag–CFM against standard CFM and Invertible Neural Network (INN) baselines on five datasets of increasing complexity: gas turbine combustor ($P{=}6$), airfoil aerodynamics ($P{=}14$), a scalable multi-objective benchmark ($P$ up to 100), thin film optical coating ($P{=}128$), and FashionMNIST image statistics ($P{=}784$). All metrics are averaged over 5 runs. Implementation details are in Appendix F.

**Evaluation Metrics.** We assess model performance using three complementary metrics:

- **Forward MSE**: Mean squared error between true labels and predicted labels via the model's forward (analysis) pass: $\mathrm{MSE} = \mathbb{E}[\|y - \hat{y}_\theta(x)\|^2]$. This measures the model's accuracy as a fast surrogate for the expensive forward simulator.
- **Round-Trip Error**: For target labels $y^*$, generate designs $x_{\mathrm{gen}}$ via the inverse pass, then evaluate through a ground-truth forward function $f$: $\mathrm{RT} = \mathbb{E}[\|y^* - f(x_{\mathrm{gen}})\|^2]$. This measures whether generated designs actually achieve the requested performance specifications. For the Gas Turbine Combustor and Unifoil datasets, $f$ is a pre-trained surrogate model; for DTLZ, thin film, and FashionMNIST, $f$ is the exact analytical mapping, enabling error-free evaluation.
- **Design Diversity**: The variance of design parameters across multiple samples generated for the same target label, averaged over design dimensions: $\mathrm{Div} = \mathbb{E}_{y^*}[\mathrm{Var}_z[x_{\mathrm{gen}}(y^*, z)]]$. Higher diversity indicates the model captures the multimodality of the solution space rather than collapsing to a single mode.

For the two datasets evaluated with learned forward surrogates, Appendix D reports additional robustness checks under surrogate-output noise and independent Unifoil surrogate architectures.

## 5.1. Gas Turbine Combustor

The gas turbine combustor dataset (Krueger et al., 2024) contains parameterized combustor geometries with $P{=}6$ design parameters and $L{=}3$ performance labels from CFD simulations (details in Appendix C). We use the augmented version of the dataset, containing 200,000 samples generated via calibrated surrogate models. These surrogates serve as the ground truth for our evaluation, having been validated against RANS CFD simulations with mean absolute errors consistently below 0.03 across all normalized performance labels. Diag-CFM, CFM and INN models all have approximately 2.1M parameters for fair comparison.

**Forward Performance.** For the forward prediction task (design $\rightarrow$ labels), Diag–CFM achieves $6.5\times$ and $12\times$ lower MSE than INN and CFM respectively, demonstrating that zero-anchoring improves both training stability (Figure 2) and predictive accuracy.

**Inverse Performance.** Diag–CFM achieves over an order of magnitude lower round-trip error than both INN and CFM, while maintaining comparable design diversity.

**Accuracy-Conditioned Diversity.** Raw diversity metrics can be misleading: a model generating inaccurate designs may exhibit high diversity simply because generated samples do not correspond to the target performance. To address this, we analyze how design diversity changes when filtering by round-trip error accuracy. Figure 3 shows diversity

*Table 1.* Performance comparison on Gas Turbine Combustor dataset. All models have approximately 2.1M parameters. Forward MSE measures prediction accuracy. Round-trip error measures MSE between target labels and surrogate-predicted labels of generated designs. Design diversity measures mean variance of design parameters across generated samples for the same target. All values over 5 runs. Best values are in bold.

| METRIC | INN | CFM | DIAG-CFM |
|---|---|---|---|
| PARAMETERS | $2.13 \times 10^6$ | $2.11 \times 10^6$ | $2.12 \times 10^6$ |
| FORWARD MSE | $9.09 \times 10^{-3}$ | $1.70 \times 10^{-2}$ | $\mathbf{1.39 \times 10^{-3}}$ |
| | $\pm 1.09 \times 10^{-3}$ | $\pm 0.39 \times 10^{-2}$ | $\pm 0.59 \times 10^{-3}$ |
| ROUND-TRIP ERROR | $1.62 \times 10^{-2}$ | $1.85 \times 10^{-2}$ | $\mathbf{1.27 \times 10^{-3}}$ |
| | $\pm 0.13 \times 10^{-2}$ | $\pm 0.23 \times 10^{-2}$ | $\pm 0.06 \times 10^{-3}$ |
| DESIGN DIVERSITY | $\mathbf{5.27 \times 10^{-2}}$ | $3.22 \times 10^{-2}$ | $2.95 \times 10^{-2}$ |
| | $\pm 0.11 \times 10^{-2}$ | $\pm 0.14 \times 10^{-2}$ | $\pm 0.07 \times 10^{-2}$ |

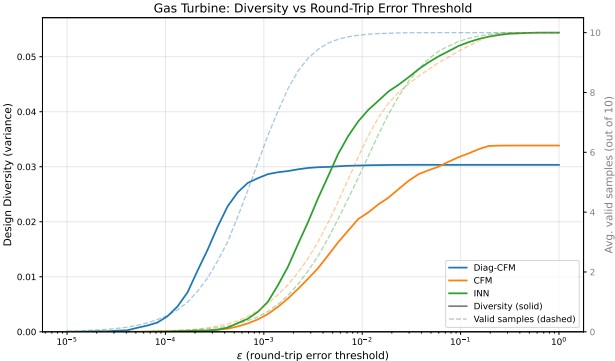

*Figure 3.* Design diversity as a function of round-trip error threshold $\varepsilon$ for the gas turbine combustor dataset. Solid lines show mean design diversity (variance) computed only over samples with error $< \varepsilon$; dashed lines show the average number of valid samples. Diag–CFM (blue) maintains high diversity at strict accuracy thresholds where INN and CFM have few valid samples, demonstrating that its designs are both accurate and diverse.

as a function of the error threshold $\varepsilon$—only designs with round-trip error below $\varepsilon$ contribute to the diversity computation. At strict thresholds ($\varepsilon < 10^{-2}$), Diag–CFM maintains substantial diversity while INN and CFM have few or no valid samples. As the threshold relaxes, INN's diversity increases but only because it includes less accurate designs. This analysis reveals that Diag–CFM's lower raw diversity score reflects its concentration on accurate solutions rather than a limitation in exploration capability.

## 5.2. Unifoil

The Unifoil dataset ([Kanchi et al., 2025](#)) contains RANS simulations of airfoils parameterized by a $P{=}14$-dimensional POD basis, with $L{=}3$ aerodynamic coefficients as labels, conditioned on angle of attack and Mach number (details in Appendix C). For round-trip error evaluation, we trained a neural network surrogate mapping design and physical parameters to aerodynamic coefficients, achieving validation MSE of $2.5 \times 10^{-5}$. Diag-CFM, CFM and INN

*Table 2.* Performance comparison on Unifoil dataset. All models have approximately 2.1M parameters. Forward MSE measures prediction accuracy. Round-trip error measures MSE between target labels and surrogate-predicted labels of generated designs. Design diversity measures mean variance of design parameters across generated samples for the same target. All values over 5 runs. Best values are in bold.

| METRIC | INN | CFM | DIAG-CFM |
|---|---|---|---|
| PARAMETERS | $2.18 \times 10^6$ | $2.13 \times 10^6$ | $2.14 \times 10^6$ |
| FORWARD MSE | $4.13 \times 10^{-3}$ | $9.66 \times 10^{-3}$ | $\mathbf{1.43 \times 10^{-3}}$ |
| | $\pm 1.29 \times 10^{-3}$ | $\pm 3.08 \times 10^{-3}$ | $\pm 0.54 \times 10^{-3}$ |
| ROUND-TRIP ERROR | $3.82 \times 10^{-3}$ | $1.10 \times 10^{-2}$ | $\mathbf{1.68 \times 10^{-3}}$ |
| | $\pm 1.93 \times 10^{-3}$ | $\pm 0.25 \times 10^{-2}$ | $\pm 0.94 \times 10^{-3}$ |
| DESIGN DIVERSITY | $\mathbf{1.44 \times 10^{-2}}$ | $5.35 \times 10^{-3}$ | $1.71 \times 10^{-3}$ |
| | $\pm 2.19 \times 10^{-2}$ | $\pm 0.69 \times 10^{-3}$ | $\pm 0.52 \times 10^{-3}$ |

models all have approximately 2.1M parameters for fair comparison.

**Forward Performance.** Diag–CFM achieves $2.9\times$ and $6.8\times$ lower forward MSE than INN and CFM respectively, confirming that zero-anchoring benefits extend to conditional generation with physical parameters (angle of attack, Mach number).

**Inverse Performance.** Diag–CFM achieves the lowest round-trip error, outperforming INN by $2.3\times$ and CFM by $6.5\times$. CFM generates the most diverse designs, though accuracy-conditioned analysis (Appendix G) shows Diag–CFM maintains competitive diversity at strict accuracy thresholds.

## 5.3. DTLZ Benchmark

To evaluate scalability beyond the gas turbine ($P{=}6$) and Unifoil ($P{=}14$) datasets, we employ the DTLZ2 test function ([Deb et al., 2005](#)), a multi-objective optimization benchmark where design dimension $P$ can be scaled arbitrarily. Crucially, the forward mapping is analytical, enabling exact round-trip error computation without surrogate model approximation. We fix $L{=}3$ objectives and vary $P \in \{12, 24, 50, 100\}$ (details in Appendix C).

**Forward Performance.** For the forward prediction task (design $\rightarrow$ objectives), Diag–CFM achieves the lowest mean squared error at lower dimensions ($P{=}12$: $1.69 \times 10^{-3}$, $P{=}24$: $3.62 \times 10^{-3}$), outperforming both INN and CFM by an order of magnitude. At higher dimensions ($P{=}50, 100$), INN achieves slightly lower forward MSE ($\sim 2 \times 10^{-3}$) than Diag–CFM, which may be a product of the nature of this dataset. Standard CFM consistently shows the worst forward performance across all dimensions, with MSE remaining above $2.8 \times 10^{-2}$.

**Inverse Performance.** Critically, for the inverse design task, Diag–CFM achieves substantially lower round-trip error across all dimensions, demonstrating superior in-

*Table 3.* DTLZ2 benchmark results comparing INN, CFM, and Diag-CFM across design dimensions ($L$=3 objectives fixed). Forward MSE measures prediction accuracy (design → labels). Round-trip error measures inverse design quality using the analytical ground truth. All values over 5 runs. Best values per dimension are in bold.

| P | INN | CFM | Diag-CFM |
|---|---|---|---|
| | | FORWARD MSE | |
| 12 | $2.01 \times 10^{-2}$ $\pm 0.06 \times 10^{-2}$ | $2.87 \times 10^{-2}$ $\pm 0.17 \times 10^{-2}$ | $\mathbf{1.69 \times 10^{-3}}$ $\pm 0.17 \times 10^{-3}$ |
| 24 | $4.58 \times 10^{-3}$ $\pm 0.29 \times 10^{-3}$ | $3.28 \times 10^{-2}$ $\pm 0.14 \times 10^{-2}$ | $\mathbf{3.62 \times 10^{-3}}$ $\pm 0.51 \times 10^{-3}$ |
| 50 | $\mathbf{2.10 \times 10^{-3}}$ $\pm 0.82 \times 10^{-3}$ | $3.36 \times 10^{-2}$ $\pm 0.20 \times 10^{-2}$ | $6.83 \times 10^{-3}$ $\pm 1.83 \times 10^{-3}$ |
| 100 | $\mathbf{1.22 \times 10^{-3}}$ $\pm 0.21 \times 10^{-3}$ | $4.17 \times 10^{-2}$ $\pm 0.72 \times 10^{-2}$ | $9.58 \times 10^{-3}$ $\pm 1.15 \times 10^{-3}$ |
| | | ROUND-TRIP ERROR | |
| 12 | $3.76 \times 10^{-2}$ $\pm 0.26 \times 10^{-2}$ | $3.20 \times 10^{-2}$ $\pm 0.60 \times 10^{-2}$ | $\mathbf{9.10 \times 10^{-4}}$ $\pm 0.79 \times 10^{-4}$ |
| 24 | $7.96 \times 10^{-2}$ $\pm 0.22 \times 10^{-2}$ | $3.39 \times 10^{-2}$ $\pm 0.36 \times 10^{-2}$ | $\mathbf{1.44 \times 10^{-3}}$ $\pm 0.06 \times 10^{-3}$ |
| 50 | $9.39 \times 10^{-2}$ $\pm 0.36 \times 10^{-2}$ | $3.58 \times 10^{-2}$ $\pm 0.47 \times 10^{-2}$ | $\mathbf{2.62 \times 10^{-3}}$ $\pm 0.25 \times 10^{-3}$ |
| 100 | $9.55 \times 10^{-2}$ $\pm 0.19 \times 10^{-2}$ | $4.52 \times 10^{-2}$ $\pm 0.91 \times 10^{-2}$ | $\mathbf{5.26 \times 10^{-3}}$ $\pm 0.71 \times 10^{-3}$ |

*Table 4.* Thin Film Optical Coating benchmark results ($P$=128, $L$=64). Forward MSE measures prediction accuracy for reflectance spectra. Round-trip error measures inverse design quality using the exact TMM forward map. All values are mean $\pm$ standard deviation over 5 runs. Best values are in bold.

| METHOD | PARAMETERS | FORWARD MSE | ROUND-TRIP ERROR |
|---|---|---|---|
| DIAG-CFM | $3.54 \times 10^6$ | $\mathbf{1.104 \times 10^{-2}}$ $\pm 0.001 \times 10^{-2}$ | $\mathbf{2.247 \times 10^{-2}}$ $\pm 0.032 \times 10^{-2}$ |
| CFM | $3.41 \times 10^6$ | $1.905 \times 10^{-2}$ $\pm 0.009 \times 10^{-2}$ | $2.902 \times 10^{-2}$ $\pm 0.026 \times 10^{-2}$ |
| INN | $3.69 \times 10^6$ | $2.384 \times 10^{-2}$ $\pm 0.020 \times 10^{-2}$ | $3.114 \times 10^{-2}$ $\pm 0.015 \times 10^{-2}$ |

*Table 5.* FashionMNIST image-statistics benchmark results ($P$=784, $L$=5). Forward MSE measures prediction accuracy for normalized nonlinear image statistics. Round-trip error measures inverse quality using the exact analytical image-statistics forward map. All values are mean $\pm$ standard deviation over 5 runs. Best values are in bold.

| METHOD | PARAMETERS | FORWARD MSE | ROUND-TRIP ERROR |
|---|---|---|---|
| DIAG-CFM | $4.31 \times 10^6$ | $\mathbf{7.20 \times 10^{-4}}$ $\pm 1.18 \times 10^{-4}$ | $\mathbf{9.15 \times 10^{-3}}$ $\pm 2.64 \times 10^{-3}$ |
| CFM | $4.24 \times 10^6$ | $2.56 \times 10^{-2}$ $\pm 4.72 \times 10^{-3}$ | $1.35 \times 10^{-2}$ $\pm 5.57 \times 10^{-3}$ |
| INN | $5.33 \times 10^6$ | $1.77 \times 10^{-3}$ $\pm 3.46 \times 10^{-4}$ | $2.73 \times 10^{-2}$ $\pm 3.19 \times 10^{-3}$ |

verse mapping quality. At $P$=12, Diag–CFM achieves $9.10 \times 10^{-4}$, outperforming CFM ($3.20 \times 10^{-2}$) by $35\times$ and INN ($3.76 \times 10^{-2}$) by $42\times$. This advantage persists at high dimensions: at $P$=100, Diag–CFM achieves $5.26 \times 10^{-3}$ compared to CFM ($4.52 \times 10^{-2}$) and INN ($9.55 \times 10^{-2}$). The increasing round-trip error for INN at high dimensions ($\sim 10^{-1}$ at $P$=100) suggests that while INN can learn a good forward mapping, its inverse mapping degrades significantly with dimension. This highlights Diag–CFM's strength in maintaining bidirectional consistency, which is essential for generative design applications.

**Design Diversity.** Raw diversity scores are comparable across all three models, with INN showing slightly higher values (see Table 10 in Appendix G). However, accuracy-conditioned analysis (Appendix G) reveals that Diag–CFM reaches full valid samples at substantially lower $\varepsilon$ values than CFM or INN across all dimensions, confirming that its designs are both accurate and diverse—whereas the higher raw diversity of competing methods largely stems from inaccurate samples.

### 5.4. Thin Film Optical Coating

To further test scalability with exact ground truth, we add a thin film optical coating benchmark, a canonical inverse problem in photonics (Ma et al., 2025). Designs are stacks of $P$=128 alternating dielectric layers ($SiO_2$/$TiO_2$) with thicknesses in $[10, 300]$ nm, and labels are $L$=64 reflectance values over wavelengths from 400–800 nm. The forward map is the Transfer Matrix Method (TMM) (Luce et al.,

2022), an analytical computation involving trigonometric phase accumulation, matrix chain multiplication across layers, and modulus-squared reflectance. Round-trip error is therefore evaluated without a learned surrogate.

Diag–CFM achieves the lowest forward MSE and round-trip error. Compared with CFM, it reduces forward MSE by $1.7\times$ and round-trip error by $1.3\times$; compared with INN, the reductions are $2.2\times$ and $1.4\times$. This benchmark is higher-dimensional than the DTLZ settings where INN has the best forward MSE, yet Diag–CFM is best in both directions, suggesting that the DTLZ forward-prediction exception is not a generic high-dimensional failure mode.

### 5.5. FashionMNIST Image Statistics

To test whether the method extends beyond MLP-based parametric design spaces, we add a high-dimensional image benchmark based on FashionMNIST (Xiao et al., 2017). Each design is a $28 \times 28$ grayscale garment image with $P$=784 pixel degrees of freedom. Labels are $L$=5 exact nonlinear image statistics: mean intensity, gradient energy, variance of local patch variances, pixel kurtosis, and log contrast. The inverse task is many-to-one: many distinct garment images can share the same image-statistics vector. Unlike the previous MLP experiments, Diag–CFM and CFM use a CNN velocity network with U-Net-style encoder-decoder structure and bottleneck self-attention (details in Appendix F).

Diag–CFM achieves the best forward and inverse performance at $P{=}784$. Relative to CFM, it reduces forward MSE by $36\times$ and round-trip error by $1.5\times$; relative to INN, it reduces forward MSE by $2.5\times$ and round-trip error by $3.0\times$. This extends the empirical evidence beyond low-dimensional vector-valued design spaces: the same ranking holds for a pixel-space inverse problem using a spatial CNN architecture and exact analytical labels.

### 5.6. Uncertainty Quantification

We evaluate our four uncertainty metrics on three practical tasks. Full results are presented in Appendix H.

**Select-Best.** For selecting the best candidate among $K{=}10$ generations per target, the Diag–CFM specific metrics (Zero-Deviation, Self-Consistency) consistently outperform baseline methods, achieving 5–31% improvement over random selection. Baseline metrics (Ensemble Variance, FM Loss) often perform *worse* than random selection due to weak or negative correlation with generation quality.

**Error-Rejection.** In deployment, it is often preferable to abstain from a prediction rather than return an unreliable design. We test whether uncertainty metrics can identify which generations to reject: for each of 1,000 target labels, we generate one design, rank all samples by uncertainty, and progressively reject the most uncertain fraction. A useful metric should yield lower mean error on the retained samples as the rejection rate increases. At 20% rejection, Zero-Deviation achieves 8–27% error reduction across datasets, with Self-Consistency performing comparably. FM Loss provides minimal improvement (3–7%), with Ensemble Variance even increasing error at high dimensions.

**Out-of-Distribution Detection.** We generate OOD targets by sampling label-space points whose nearest training neighbor lies at 2–8% of each dimension's range—close enough to the boundary to make detection challenging. Zero-Deviation consistently achieves the best AUC scores: 0.96 (Unifoil), 0.73–0.82 (DTLZ), and 0.75 (Gas Turbine). Self-Consistency follows with AUCs of 0.87, 0.51–0.77, and 0.69 respectively. Baseline metrics show mixed results: on Unifoil, FM Loss (0.89) performs well, but on Gas Turbine and DTLZ it achieves only 0.59–0.69. Ensemble Variance is consistently the weakest (0.62–0.85). That Zero-Deviation matches or exceeds all alternatives is particularly significant as it is always computed as a byproduct of generation. Appendix H.3 adds physically grounded Unifoil OOD tests where Zero-Deviation exceeds 0.98 AUC on negative drag, extreme lift-to-drag ratio, below-drag-polar targets, and extrapolated Mach/AoA conditioning shifts. Appendix E shows that this signal is stable across a range of ODE solvers.

## 6. Conclusion

We introduced Diagonal Flow Matching (Diag–CFM), a zero-anchoring strategy for conditional flow matching that yields a permutation-invariant learning problem with respect to design and label coordinates. This simple modification eliminates the sensitivity to coordinate ordering present in standard CFM, yielding substantially improved round-trip accuracy across all tested datasets. Crucially, this accuracy does not come at the cost of mode collapse: we demonstrated that Diag–CFM maintains high design diversity even at strict error thresholds where baseline models fail to produce valid candidates.

For uncertainty quantification, we developed two architecture-specific metrics—Zero-Deviation and Self-Consistency—that consistently outperform ensemble-based and flow matching loss alternatives. Zero-Deviation is particularly attractive as it is computed as a byproduct of generation, offering reliable epistemic uncertainty estimation without the computational overhead of training multiple ensemble members.

These contributions enable a key capability for deploying generative design in engineering practice: the ability to flag uncertain predictions rather than silently producing unreliable designs. This is especially valuable when downstream decisions (e.g. manufacturing, deployment) are costly.

**Limitations.** While our UQ metrics substantially improve upon baselines, the gap to oracle performance (34–57% for error-rejection) indicates room for improvement. Additionally, while Diag–CFM excels at inverse generation, we observed that for forward prediction on DTLZ at high dimensions ($P{\geq}50$), standard INN baselines remain competitive. Future work might explore hybrid architectures to capture the strengths of both approaches.

**Future Work.** Promising directions include integrating Diag–CFM into active learning loops where uncertainty guides data acquisition to resolve high-error regions efficiently. Finally, given the method's effective scaling to high-dimensional spaces, a natural next step is to apply Diag–CFM to inverse problems with larger parameter sets, such as topology optimization or molecular discovery.

## Acknowledgements

M.C. acknowledges financial support from the German Federal Ministry for Economic Affairs through grant no. 03EE5186B and from Siemens Energy. H.G. acknowledges financial support from the German Research Foundation (DFG) through SPP 2403 "Carnot Batteries".

## Impact Statement

This paper presents work whose goal is to advance the field of Machine Learning. There are many potential societal consequences of our work, none which we feel must be specifically highlighted here.

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

## A. Proofs

*Proof of Proposition 4.1.* Let $\Pi_P \in \mathbb{R}^{P,P}$ and $\Pi_L \in \mathbb{R}^{L,L}$ be permutation matrices, and $\Pi := \text{diag}(\Pi_P, \Pi_L) = \begin{bmatrix} \Pi_P & 0 \\ 0 & \Pi_L \end{bmatrix}$, then

$$\begin{bmatrix} \Pi_P X - Z \\ -\Pi_L Y \end{bmatrix} = \begin{bmatrix} \Pi_P(X - \Pi_P^{-1} Z) \\ -\Pi_L Y \end{bmatrix} = \Pi \begin{bmatrix} X - \Pi_P^{-1} Z \\ -Y \end{bmatrix}.$$

Because the entries of $Z$ are i.i.d., we have that $\begin{bmatrix} \Pi_P X - Z \\ -\Pi_L Y \end{bmatrix}$ and $\Pi \begin{bmatrix} X - Z \\ -Y \end{bmatrix}$ are equal in law. $\qquad \square$

*Proof of Proposition 4.2.* To see this consider as an example $X$ with absolutely continuous i.i.d. entries, and $Y := [X]_{P-L+1:P}$. Then

$$U = \begin{bmatrix} [X]_{1:P-L} - Z \\ [X]_{P-L+1:P} - Y \end{bmatrix} = \begin{bmatrix} [X]_{1:P-L} - Z \\ 0 \end{bmatrix},$$

has $L$ entries which are almost surely 0, while for $\Pi_P$ permuting the first with the last entry,

$$U = \begin{bmatrix} [\Pi_P X]_{1:P-L} - Z \\ [\Pi_P X]_{P-L+1:P} - Y \end{bmatrix}$$

only has $L - 1$ almost surely 0 entries. $\qquad \square$

## B. CFM vs Diag–CFM Ablation

We compare standard CFM and Diag–CFM under four different parameter orderings on the gas turbine combustor dataset, following the training protocol described in Appendix F. For each ordering, we train 5 models with different random seeds and report the mean and standard deviation of the final round-trip error. Table 6 provides numerical values for the ablation study visualized in Figure 2.

We observe that standard CFM performance is highly dependent on the ordering of parameters, with mean final round-trip error varying by nearly an order of magnitude across orderings ($4.66 \times 10^{-3}$ to $3.02 \times 10^{-2}$). In contrast, Diag–CFM achieves consistent performance ($\sim 1.35 \times 10^{-3}$) regardless of parameter ordering, confirming the permutation-invariance of the learning problem implied by Proposition 4.1. Moreover, Diag–CFM exhibits substantially lower run-to-run variance *within* each ordering: its standard deviations range from the order of $10^{-4}$ to $10^{-5}$, whereas CFM standard deviations range from $10^{-4}$ to $10^{-3}$, indicating that Diag–CFM training is more stable across random initializations as well. We also note, that the gas turbine dataset is already normalized to $[0, 1]$, so the observed sensitivity is attributable purely to coordinate ordering; on datasets with heterogeneous scales, the effect would likely be more pronounced.

*Table 6.* End of training round-trip error (mean $\pm$ std over 5 runs) for gas turbine dataset with different parameter orderings.

| ROUND-TRIP ERROR | CFM | DIAG–CFM |
|---|---|---|
| PARAMETER ORDERING 1 | $(1.93 \pm 0.29) \times 10^{-2}$ | $(1.38 \pm 0.12) \times 10^{-3}$ |
| PARAMETER ORDERING 2 | $(2.68 \pm 0.59) \times 10^{-2}$ | $(1.31 \pm 0.07) \times 10^{-3}$ |
| PARAMETER ORDERING 3 | $(4.66 \pm 0.38) \times 10^{-3}$ | $(1.31 \pm 0.10) \times 10^{-3}$ |
| PARAMETER ORDERING 4 | $(3.02 \pm 0.15) \times 10^{-2}$ | $(1.38 \pm 0.15) \times 10^{-3}$ |

## C. Dataset Descriptions

### C.1. Gas Turbine Combustor

We use an augmented version of the gas turbine combustor dataset from (Krueger et al., 2024), who developed a generative design methodology for premixed combustion systems. Each sample is a parameterized combustor (plenum, premixing tube, combustion chamber with central fuel lance), simulated via steady RANS CFD and an acoustic network model to obtain labels.

The six design parameters are $x = (R_A,\ N_H,\ D_M,\ R_D,\ R_L,\ L_P)$, where $R_A$ is the ratio of free area at the vortex generators to the premixing-tube cross-section, $N_H$ the number of fuel-injection holes on the lance, $D_M$ the premixing-tube

diameter, $R_D$ the ratio of lance diameter to premixing-tube diameter, $R_L$ the premixing-tube length-to-diameter ratio, and $L_P$ the combustor plenum length (typical ranges: $R_A \in [0.63, 0.83]$, $N_H \in [2, 10]$, $D_M \in [20, 45]$ mm, $R_D \in [0.35, 0.55]$, $R_L \in [4, 12]$, $L_P \in [200, 900]$ mm).

The three performance labels are $y = (U_M, \ \Delta p_{t,\text{rel}}, \ G)$: $U_M$ is the fuel/air unmixedness at the premixing-tube outlet (a mass-flow-weighted normalized standard deviation of mixture fraction), $\Delta p_{t,\text{rel}}$ is the normalized total pressure loss between domain inlet and outlet, and $G$ is the thermoacoustic growth rate, derived from the acoustic network (GENEAC) eigenanalysis.

## C.2. Unifoil

The Unifoil dataset (Kanchi et al., 2025) is a comprehensive, publicly available airfoil dataset based on Reynolds-averaged Navier-Stokes (RANS) simulations. The dataset contains over 500,000 aerodynamic simulations spanning a wide range of flow conditions, including both laminar-turbulent transitional flows and fully turbulent flows across incompressible to compressible flow regimes.

We work with the fully turbulent subsection of the dataset. Each sample corresponds to a RANS simulation of an airfoil under a given angle of attack and Mach number. Each airfoil is parametrized using a $P{=}14$-dimensional basis obtained via proper orthogonal decomposition (POD) from a diverse collection of airfoils. As performance labels we use the $L{=}3$ values precomputed by the authors for the lift, drag, and moment coefficients. We refer to (Kanchi et al., 2025) for more details.

For round-trip error evaluation, we trained a neural network surrogate to serve as ground truth. The surrogate is an MLP with 5 hidden layers of width 512, LeakyReLU activations, and layer normalization. It maps the 16-dimensional input (14 design parameters concatenated with angle of attack and Mach number) to the 3 aerodynamic coefficients. Training used the Adam optimizer with learning rate $10^{-3}$, batch size 256, and a ReduceLROnPlateau scheduler. The surrogate achieves a validation MSE of $2.5 \times 10^{-5}$.

Figure 4 shows airfoil geometries from the dataset alongside airfoils generated by Diag–CFM. The generated airfoils exhibit realistic and diverse shapes consistent with the dataset distribution.

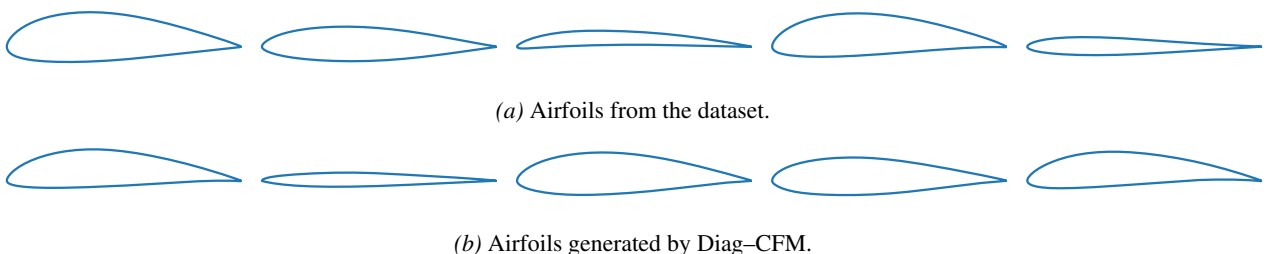

*(a)* Airfoils from the dataset.

*(b)* Airfoils generated by Diag–CFM.

*Figure 4.* Airfoil geometries reconstructed from the 14-dimensional POD basis. (a) Airfoils from the Unifoil dataset. (b) Airfoils generated by Diag–CFM from randomly sampled performance labels and physical conditioning parameters (angle of attack, Mach number). The generated shapes are realistic and diverse, reflecting the model's ability to produce plausible designs for given aerodynamic targets.

## C.3. DTLZ Benchmark

The DTLZ2 function (Deb et al., 2005) is a widely-used benchmark in multi-objective optimization that offers two key advantages: (1) the design dimension $P$ can be scaled arbitrarily, enabling systematic study of high-dimensional behavior, and (2) the forward mapping from designs to objectives is analytical, allowing exact round-trip error computation without surrogate model approximation.

The DTLZ2 function maps design parameters $x \in [0, 1]^P$ to $L$ objectives. For a design $x = (x_1, \ldots, x_P)$ with $P \geq L$, let

$g(x_M) = \sum_{i=L}^{P}(x_i - 0.5)^2$ where $x_M = (x_L, \ldots, x_P)$ are the "distance" parameters. The objectives are:

$$f_1(x) = (1 + g) \prod_{i=1}^{L-1} \cos\left(\frac{\pi}{2} x_i\right),$$

$$f_m(x) = (1 + g) \sin\left(\frac{\pi}{2} x_{L-m}\right) \prod_{i=1}^{L-1-m} \cos\left(\frac{\pi}{2} x_i\right), \quad 2 \le m \le L.$$

The Pareto-optimal front is the first quadrant of a unit hypersphere when $g = 0$ (i.e., $x_i = 0.5$ for $i \ge L$).

## C.4. Thin Film Optical Coating

The thin film benchmark models normal-incidence reflection from a multilayer optical coating. Each design $x \in [0, 1]^P$ specifies layer thicknesses, which are mapped to physical thicknesses in $[10, 300]$ nm. Layers alternate between $SiO_2$ ($n=1.45$) and $TiO_2$ ($n=2.30$) on a glass substrate ($n_{sub}=1.5$), with air as the incident medium ($n_0=1.0$). We use $P=128$ layers and evaluate reflectance at $L=64$ uniformly spaced wavelengths between 400 and 800 nm.

For each wavelength $\lambda$, the phase thickness of layer $k$ is $\delta_k(\lambda) = 2\pi n_k d_k / \lambda$, and the layer transfer matrix is

$$M_k(\lambda) = \begin{bmatrix} \cos\delta_k & -i\sin\delta_k/n_k \\ -in_k\sin\delta_k & \cos\delta_k \end{bmatrix}.$$

Multiplying the $P$ layer matrices gives the system matrix $M(\lambda) = \prod_{k=1}^{P} M_k(\lambda)$. The reflection coefficient is computed from $M(\lambda)$, $n_0$, and $n_{sub}$, and the label is the reflectance $R(\lambda) = |r(\lambda)|^2$. This exact analytical forward map is highly nonlinear because the phase terms are periodic in layer thickness, the transfer matrices compound across layers, and the modulus squared discards phase information.

We generate 50,000 training samples, 10,000 validation samples, and 10,000 test samples using fixed random seeds. Design parameters are sampled uniformly in $[0, 1]$ and labels are normalized to $[0, 1]$ using training-set per-wavelength min/max statistics. For round-trip evaluation, generated designs are clamped to $[0, 1]$ and evaluated with the same TMM forward map; no learned surrogate is used.

## C.5. FashionMNIST Image Statistics

The FashionMNIST benchmark uses the standard FashionMNIST garment images (Xiao et al., 2017) as designs. Each design is represented as an image tensor $I \in [0, 1]^{1 \times 28 \times 28}$, corresponding to $P=784$ pixel degrees of freedom. Instead of using class labels, we define an exact analytical forward map from images to $L=5$ normalized image statistics: mean intensity, gradient energy, variance of local $3 \times 3$ patch variances, pixel kurtosis, and log contrast. Table 7 lists the five statistics.

*Table 7.* FashionMNIST image-statistics labels. Let $\Omega$ be the $28 \times 28$ pixel grid, $N = |\Omega|$, $\mu = N^{-1}\sum_{u\in\Omega} I_u$, and $s^2 = (N-1)^{-1}\sum_{u\in\Omega}(I_u - \mu)^2$. The notation $\mathrm{Var}^{(1)}$ denotes sample variance, matching the implementation. The $3 \times 3$ neighborhoods $\mathcal{N}_3(u)$ use zero padding at image boundaries. The sets $B_{100}$ and $D_{100}$ contain the 100 brightest and 100 darkest pixels, respectively.

| Statistic | Formula |
| --- | --- |
| Mean intensity | $\frac{1}{N}\sum_{u\in\Omega} I_u$ |
| Gradient energy | $\frac{1}{2}\left(\mathrm{mean}_{i,j}(I_{i+1,j} - I_{i,j})^2 + \mathrm{mean}_{i,j}(I_{i,j+1} - I_{i,j})^2\right)$ |
| Local-variance variance | $\mathrm{Var}^{(1)}_{u\in\Omega}\left(\mathrm{Var}^{(1)}_{v\in\mathcal{N}_3(u)} I_v\right)$ |
| Pixel kurtosis | $\frac{1}{N}\sum_{u\in\Omega}\frac{(I_u - \mu)^4}{\max(s, 10^{-8})^4}$ |
| Log contrast | $\log\left(\frac{\frac{1}{100}\sum_{u\in B_{100}} I_u}{\max\left(\frac{1}{100}\sum_{u\in D_{100}} I_u, 10^{-4}\right)}\right)$ |

The gradient-energy label is the average squared finite difference over horizontal and vertical neighbors. The local-variance label computes the sample variance across all zero-padded $3 \times 3$ patch variances. Pixel kurtosis and log contrast use the numerical clamps shown in Table 7. Labels are normalized to $[0, 1]$ using training-set min/max statistics.

We use 50,000 training images, 10,000 validation images, and 10,000 test images. For round-trip evaluation, generated images are clamped to $[0, 1]$ and evaluated with the same analytical image-statistics map; no learned surrogate or classifier is used.

## D. Surrogate Robustness

Gas turbine combustor and Unifoil round-trip evaluation use learned forward surrogates because querying the original high-fidelity simulators for every generated design is impractical. We therefore check whether the method ranking is sensitive to surrogate perturbations or to the particular surrogate architecture.

First, we inject Gaussian noise into surrogate outputs at levels from 1% to 20% of the per-label training-set standard deviation and recompute round-trip errors. Table 8 reports the endpoints of this sweep. The ranking Diag–CFM < INN < CFM is unchanged for both datasets, even at 20% output noise.

Second, for Unifoil we train nine independent surrogates: three architectures with different depths, widths, and activations, each with three random seeds. Their validation MSEs range from $6.9 \times 10^{-5}$ to $8.6 \times 10^{-5}$. Re-evaluating the same generated designs under all nine surrogates gives the same ranking in every case (Table 8), with inter-method gaps exceeding the variation induced by the choice of surrogate.

*Table 8.* Surrogate robustness checks for datasets whose round-trip metric uses a learned forward surrogate. For noise injection, Gaussian noise is added to surrogate outputs as a fraction of the per-label training-set standard deviation; entries are ensemble mean $\pm$ ensemble standard deviation after averaging over five noise seeds for nonzero noise. For cross-surrogate validation, entries are mean and range over nine independently trained Unifoil surrogates. Lower is better.

| CHECK | SETTING | DIAG-CFM | INN | CFM |
|---|---|---|---|---|
| GAS TURBINE NOISE | 0% | $\mathbf{1.35 \times 10^{-3}} \pm 1.11 \times 10^{-4}$ | $1.63 \times 10^{-2} \pm 1.50 \times 10^{-3}$ | $1.86 \times 10^{-2} \pm 2.36 \times 10^{-3}$ |
| GAS TURBINE NOISE | 20% | $\mathbf{2.75 \times 10^{-3}} \pm 1.10 \times 10^{-4}$ | $1.77 \times 10^{-2} \pm 1.50 \times 10^{-3}$ | $2.00 \times 10^{-2} \pm 2.37 \times 10^{-3}$ |
| UNIFOIL NOISE | 0% | $\mathbf{1.68 \times 10^{-3}} \pm 9.37 \times 10^{-4}$ | $3.79 \times 10^{-3} \pm 1.84 \times 10^{-3}$ | $1.10 \times 10^{-2} \pm 2.51 \times 10^{-3}$ |
| UNIFOIL NOISE | 20% | $\mathbf{3.19 \times 10^{-3}} \pm 9.37 \times 10^{-4}$ | $5.30 \times 10^{-3} \pm 1.84 \times 10^{-3}$ | $1.25 \times 10^{-2} \pm 2.51 \times 10^{-3}$ |
| UNIFOIL 9-SURROGATE CV | MEAN [RANGE] | $\mathbf{1.75 \times 10^{-3}}$ $[1.67, 1.84] \times 10^{-3}$ | $4.05 \times 10^{-3}$ $[3.11, 6.35] \times 10^{-3}$ | $1.10 \times 10^{-2}$ $[1.09, 1.10] \times 10^{-2}$ |

## E. ODE Solver Sensitivity

Our default CFM and Diag–CFM evaluations use an explicit Euler solver with 30 uniform steps. To assess the sensitivity of the conclusions to this choice, we re-evaluate the same gas turbine checkpoints with Euler steps $\{20, 30, 50, 100\}$, RK4 steps $\{20, 30, 50, 100\}$, and adaptive Dopri5 with `atol=rtol=10`$^{-5}$. The INN baseline has no numerical-integration step, so the table below focuses on the ODE-based methods and UQ scores.

*Table 9.* Gas turbine ODE solver sensitivity across Euler steps $\{20, 30, 50, 100\}$, RK4 steps $\{20, 30, 50, 100\}$, and adaptive Dopri5. Entries report the range across the nine solver configurations and the relative spread $(\max - \min)/\min$. Only ODE-based models are listed for accuracy metrics; INN has no numerical-integration step.

| GROUP | QUANTITY | RANGE | SPREAD |
|---|---|---|---|
| ACCURACY | DIAG–CFM FORWARD MSE | $1.55 \times 10^{-3}$–$1.68 \times 10^{-3}$ | 8.1% |
| ACCURACY | CFM FORWARD MSE | $1.54 \times 10^{-2}$–$2.05 \times 10^{-2}$ | 32.7% |
| ACCURACY | DIAG–CFM ROUND-TRIP | $1.31 \times 10^{-3}$–$1.42 \times 10^{-3}$ | 8.3% |
| ACCURACY | CFM ROUND-TRIP | $1.68 \times 10^{-2}$–$2.49 \times 10^{-2}$ | 48.4% |
| UQ | ZERO-DEVIATION AUC | 0.772–0.774 | 0.3% |
| UQ | SELF-CONSISTENCY AUC | 0.690–0.701 | 1.6% |
| UQ | ENSEMBLE VARIANCE AUC | 0.609–0.617 | 1.3% |
| UQ | FM LOSS AUC | 0.671–0.674 | 0.5% |

Across all nine solver configurations, Diag–CFM remains lower-error than CFM. The full sweep also shows lower solver sensitivity for Diag–CFM than for CFM: Diag–CFM varies by 8.1% in forward MSE and 8.3% in round-trip error, whereas CFM varies by 32.7% and 48.4%, respectively. RK4 already stabilizes at 20 steps and agrees closely with adaptive Dopri5.

The uncertainty results are similarly stable. On the hard gas-turbine OOD split, Zero-Deviation AUC ranges from 0.7716 to 0.7742 across the nine solver configurations, while Self-Consistency ranges from 0.6899 to 0.7008, Ensemble Variance from 0.6091 to 0.6171, and FM Loss from 0.6706 to 0.6739. The ranking Zero-Deviation > Self-Consistency > FM Loss > Ensemble Variance is unchanged throughout the sweep. Thus, the discriminative signal in Zero-Deviation is preserved under solver refinement, supporting its interpretation as an endpoint-geometry signal induced by zero-anchoring.

## F. Implementation Details

### F.1. Training Algorithm

Each iteration samples a minibatch $\{(x_i, y_i)\}_{i=1}^{B}$ from the dataset, draws $z_i$ from a uniform $[0, 1]$ base distribution. We build the Diag–CFM states

$$s_{0,i}^{\mathrm{diag}} = \big[z_i;\ y_i\big], \qquad s_{1,i}^{\mathrm{diag}} = \big[x_i;\ 0_L\big],$$

sample $t_i \sim \mathcal{U}[0, 1]$, form the interpolation $s_{t,i}^{\mathrm{diag}} = (1 - t_i)\, s_{0,i}^{\mathrm{diag}} + t_i\, s_{1,i}^{\mathrm{diag}}$ with target velocity $u_i^{\mathrm{diag}} = s_{1,i}^{\mathrm{diag}} - s_{0,i}^{\mathrm{diag}} = \big[x_i - z_i;\ -y_i\big]$, and minimize the Diag–CFM loss with respect to $\theta$.

For synthesis and analysis, we solve the induced ODE using an explicit Euler integrator with 30 uniform steps on $t \in [0, 1]$, integrating forward for synthesis ($0 \to 1$) and reversing the sign for analysis ($1 \to 0$). The choice of an explicit Euler integrator with 30 uniform steps for ODE integration is standard in the flow matching literature, where the linear interpolation paths yield approximately straight trajectories that require substantially fewer integration steps than diffusion models (Lipman et al., 2023). Appendix E reports a gas turbine solver-sensitivity sweep showing stable method rankings and stable Zero-Deviation OOD AUC under Euler, RK4, and adaptive Dopri5.

### F.2. Invertible Neural Network Baseline

As an additional baseline, we train coupling-based Invertible Neural Networks (INNs) (Dinh et al., 2017; Ardizzone et al., 2019). Each INN consists of a sequence of affine coupling layers interleaved with fixed random permutations. In each coupling layer, the input is split into two halves $(u_1, u_2)$, and the transformation is

$$v_1 = u_1 \odot \exp\big(s_2(u_2)\big) + t_2(u_2),$$
$$v_2 = u_2 \odot \exp\big(s_1(v_1)\big) + t_1(v_1),$$

where $s_k, t_k$ are MLP subnets. Scale factors are soft-clamped via $s \mapsto c \cdot \tanh(s/c)$ with $c{=}2.0$ for numerical stability. The inverse is computed analytically. For the Unifoil dataset, which requires conditioning on physical parameters $c$ (angle of attack and Mach number), we use a conditional variant where each subnet receives $c$ as additional input, i.e., $s_2(u_2, c)$, $t_2(u_2, c)$, etc.

Following (Ardizzone et al., 2019) the INN is trained with a bidirectional loss

$$\mathcal{L}_{\mathrm{INN}} = \lambda_y \mathcal{L}_y + \lambda_z \mathcal{L}_z + \lambda_x \mathcal{L}_x,$$

where $\mathcal{L}_y = \|y_{\mathrm{pred}} - y\|^2$ is the forward MSE, $\mathcal{L}_z = \mathrm{MMD}^2(z_{\mathrm{pred}}, \mathcal{N}(0, I))$ encourages the latent distribution to match a standard Gaussian prior, and $\mathcal{L}_x = \mathrm{MMD}^2(x_{\mathrm{rec}}, x)$ penalizes backward reconstruction error (with $x_{\mathrm{rec}} = T^{-1}(y, z')$ for $z' \sim \mathcal{N}(0, I)$). MMD is computed with an inverse multiquadratic kernel. We use $\lambda_y{=}1$, $\lambda_z{=}1$, $\lambda_x{=}10$ across all datasets.

In all experiments, INN hyperparameters (number of coupling blocks, subnet hidden dimension, subnet depth) are tuned so that the total parameter count approximately matches that of the corresponding Diag–CFM/CFM models, ensuring fair comparison. Dataset-specific INN configurations are detailed in the following subsections.

### F.3. Gas Turbine Combustor

For the combustor dataset ($P{=}6$ design parameters and $L{=}3$ performance labels), we parameterize $v_\theta$ with an MLP consisting of 4 hidden layers of width $1024$ (hidden dimension $512 \cdot 2$), LeakyReLU activations, and no dropout. Time is injected by concatenating the scalar $t$ to the state, so the network input is $[s; t]$. Accordingly, for Diag–CFM the MLP uses input/output dimensions $(P{+}L{+}1) \to (P{+}L)$, while for the CFM baseline it uses $(P{+}1) \to P$. We train for 20 epochs with batch size 5000 and learning rate $10^{-3}$.

The INN baseline uses 4 coupling blocks with subnet hidden dimension 256, subnet depth 3, and ReLU activations ($\sim$2.1M parameters). Training uses batch size 1000, learning rate $10^{-3}$, and 50 epochs.

### F.4. Unifoil

For the Unifoil dataset ($P=14$ design parameters and $L=3$ performance labels), we parameterize $v_\theta$ with an MLP of 3 hidden layers of width 1024 (hidden dimension $512 \cdot 2$), ReLU activations, and no dropout. Unlike the gas turbine model, Unifoil requires conditioning on physical parameters (angle of attack and Mach number, $C=2$ dimensions), which are concatenated to the network input. Thus, for Diag–CFM the network maps $(P+L+C+1) \to (P+L)$, while for the CFM baseline it maps $(P+C+1) \to P$. We train for 100 epochs with batch size 100, learning rate $10^{-3}$, and a StepLR schedule (step size 20, decay factor 0.8).

The INN baseline uses the conditional variant with 4 coupling blocks, subnet hidden dimension 256, subnet depth 3, and ReLU activations ($\sim$2.2M parameters). Training uses batch size 100, learning rate $10^{-3}$, and 100 epochs.

### F.5. DTLZ Benchmark

For each dimension $P \in \{12, 24, 50, 100\}$, we generate 100,000 training samples. To evaluate scalability accurately, we mitigate the "curse of dimensionality" (where standard uniform sampling causes points to concentrate at boundaries for large $P$) using a stratified sampling strategy: the position parameters $x_1, \ldots, x_{L-1}$ are sampled uniformly from $[0, 1]$ to ensure angular coverage, while the distance parameters $x_L, \ldots, x_P$ are sampled radially around the center 0.5. Specifically, we sample a target $g$ value uniformly from $[0, g_{\max}]$ with $g_{\max} = 2$ (yielding objective values up to $\sim$3, since $f = (1+g) \cdot$ angular terms) and use a Dirichlet distribution to allocate $(x_i - 0.5)^2$ contributions across the $P-L+1$ distance coordinates. This ensures the distribution of the distance function $g(x)$ spans from optimal to sub-optimal uniformly, independent of $P$.

The MLP architecture adapts to dimension: hidden width 512 and depth 3 for $P \leq 20$; width 1024 and depth 4 for $20 < P \leq 50$; width 1024 and depth 5 for $P > 50$. Training uses batch size 1000, learning rate $10^{-3}$, and 50 epochs. We train ensembles of 5 models for both Diag–CFM and standard CFM. Evaluation uses a held-out test set of 5,000 samples generated with a different random seed.

The INN baseline architecture also scales with dimension: 2 blocks, hidden 128, subnet depth 5 for $P \leq 20$; 3 blocks, hidden 256, depth 5 for $20 < P \leq 35$; 5 blocks, hidden 224, depth 4 for $35 < P \leq 70$; 5 blocks, hidden 256, depth 4 for $P > 70$. All use LeakyReLU activations. Training uses batch size 1000, learning rate $10^{-3}$, and 50 epochs. These configurations are tuned to approximately match the Diag–CFM/CFM parameter count at each dimension.

### F.6. Thin Film Optical Coating

For the thin film benchmark, Diag–CFM and CFM use MLP velocity networks with hidden width 1024, depth 4, LeakyReLU activations, and no dropout. Diag–CFM maps $(P+L+1) \to (P+L)$, while CFM maps $(P+1) \to P$. We train for 50 epochs with batch size 1000, learning rate $10^{-3}$, and a cosine annealing learning-rate schedule with minimum learning rate $10^{-5}$.

The INN baseline uses 4 coupling blocks with subnet hidden dimension 256, subnet depth 4, and soft-clamping value 2.0, giving approximately the same parameter count as the CFM models. It is trained for 50 epochs with batch size 1000, learning rate $10^{-3}$, and the same bidirectional INN loss as the other datasets.

### F.7. FashionMNIST Image Statistics

For the FashionMNIST benchmark, Diag–CFM and CFM use the same CNN velocity-network family. The network is a U-Net-style encoder-decoder for $28 \times 28$ images with channel widths 64, 128, and 256, skip connections, FiLM conditioning from the time embedding and label state, and self-attention at the $7 \times 7$ bottleneck. Diag–CFM receives the concatenated image state, label state, and time, and predicts both image and label velocities. Standard CFM receives only image state and time, and predicts image velocity. Both models are trained for 50 epochs with batch size 256, learning rate $10^{-3}$, and a cosine annealing learning-rate schedule with minimum learning rate $10^{-5}$.

The INN baseline uses 4 affine coupling blocks with subnet hidden dimension 256, subnet depth 3, and soft-clamping value 2.0. It is trained for 50 epochs with batch size 256, learning rate $10^{-3}$, and the same bidirectional INN loss as the other

datasets.

## G. Accuracy-Conditioned Diversity

Raw diversity metrics can be misleading: a model generating inaccurate designs may exhibit high diversity simply because generated samples do not correspond to the target performance. To address this, we analyze how design diversity changes when filtering by round-trip error accuracy. The figures below show diversity as a function of the error threshold $\varepsilon$—only designs with round-trip error below $\varepsilon$ contribute to the diversity computation.

### G.1. Unifoil

Figure 5 analyzes design diversity as a function of round-trip error threshold for the Unifoil dataset. Similar to the gas turbine results (Figure 3 in the main text), Diag–CFM's valid sample count reaches maximum at lower $\varepsilon$ values than CFM, reflecting its superior inverse mapping accuracy. At strict accuracy thresholds, Diag–CFM maintains competitive diversity while CFM's apparent diversity advantage diminishes when restricted to accurate designs. INN shows intermediate behavior with moderate accuracy but lower overall diversity. This analysis demonstrates that Diag–CFM provides the best trade-off between accuracy and diversity for airfoil design.

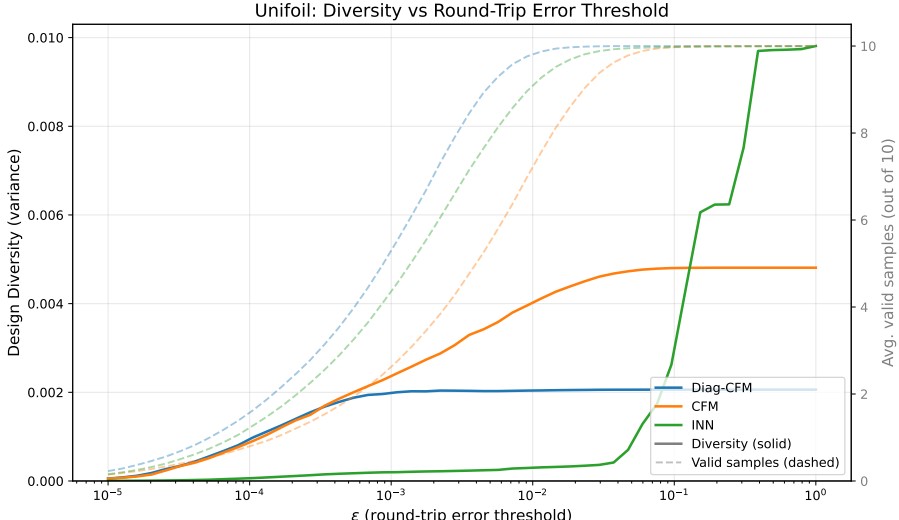

*Figure 5.* Design diversity as a function of round-trip error threshold $\varepsilon$ for the Unifoil dataset. Solid lines show mean design diversity computed only over samples with error $< \varepsilon$; dashed lines show valid sample counts. Diag–CFM achieves high diversity at strict accuracy thresholds where it has more valid samples than competing methods.

### G.2. DTLZ Benchmark

Table 10 reports raw design diversity across all dimensions. INN achieves the highest raw diversity at every $P$, followed closely by Diag–CFM, with CFM consistently lowest.

However, raw diversity can be misleading when models differ in accuracy. Figure 6 shows design diversity as a function of round-trip error threshold for the DTLZ2 benchmark at $P{=}100$, the highest-dimensional DTLZ setting we evaluate. The pattern is consistent across all tested dimensions: Diag–CFM's valid sample count (dashed blue line) reaches maximum at substantially lower $\varepsilon$ values than CFM or INN, reflecting its superior round-trip accuracy. At strict thresholds where accuracy matters, Diag–CFM maintains meaningful diversity while CFM and INN have few qualifying samples. This confirms that Diag–CFM generates designs that are both accurate and diverse, whereas the higher raw diversity of competing methods largely stems from inaccurate samples that do not satisfy the target specifications.

*Table 10.* DTLZ2 design diversity across design dimensions ($L$=3 objectives fixed). Design diversity measures mean variance of design parameters across generated samples for the same target. All values over 5 runs. Best values per dimension are in bold.

| | DESIGN DIVERSITY | | |
|---|---|---|---|
| P | INN | CFM | DIAG-CFM |
| 12 | $\mathbf{7.52 \times 10^{-2}}$ | $6.25 \times 10^{-2}$ | $7.30 \times 10^{-2}$ |
| | $\pm 0.17 \times 10^{-2}$ | $\pm 0.22 \times 10^{-2}$ | $\pm 0.10 \times 10^{-2}$ |
| 24 | $\mathbf{4.73 \times 10^{-2}}$ | $3.68 \times 10^{-2}$ | $4.03 \times 10^{-2}$ |
| | $\pm 0.23 \times 10^{-2}$ | $\pm 0.09 \times 10^{-2}$ | $\pm 0.05 \times 10^{-2}$ |
| 50 | $\mathbf{2.47 \times 10^{-2}}$ | $1.98 \times 10^{-2}$ | $1.91 \times 10^{-2}$ |
| | $\pm 0.08 \times 10^{-2}$ | $\pm 0.16 \times 10^{-2}$ | $\pm 0.06 \times 10^{-2}$ |
| 100 | $\mathbf{1.21 \times 10^{-2}}$ | $1.12 \times 10^{-2}$ | $1.05 \times 10^{-2}$ |
| | $\pm 0.04 \times 10^{-2}$ | $\pm 0.05 \times 10^{-2}$ | $\pm 0.03 \times 10^{-2}$ |

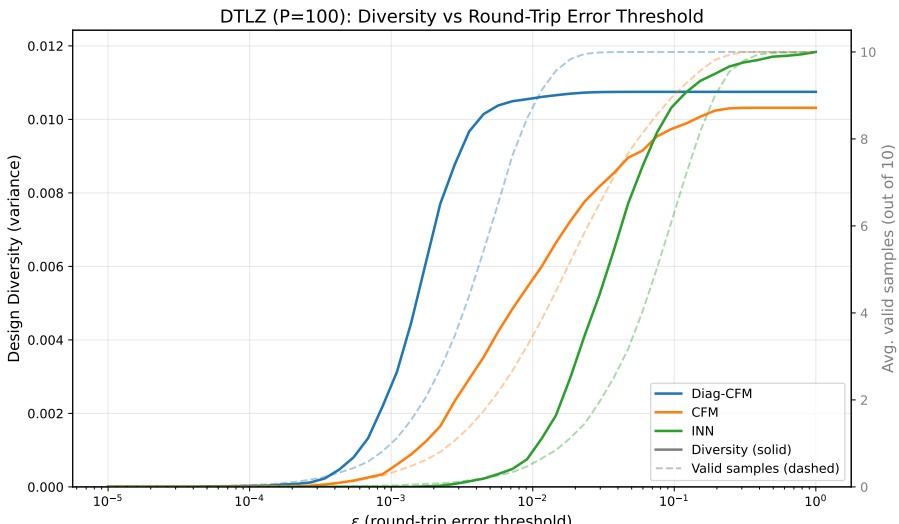

*Figure 6.* Design diversity as a function of round-trip error threshold $\varepsilon$ for the DTLZ2 benchmark ($P$=100, $L$=3). Solid lines show mean design diversity computed only over samples with error $< \varepsilon$; dashed lines show valid sample counts. Diag–CFM reaches full valid samples at lower $\varepsilon$, demonstrating superior accuracy while maintaining competitive diversity.

# H. Uncertainty Quantification Results

This appendix provides detailed results for all three uncertainty quantification tasks: select-best, error-rejection, and out-of-distribution detection.

## H.1. Select-Best

We evaluate uncertainty metrics on selecting the best candidate among $K$=10 generations for each target. For each target performance specification $y^*$, we generate candidates using different random noise seeds and select the one with lowest uncertainty score. We compare against random selection and oracle selection (choosing the candidate with lowest ground-truth error).

We evaluate four metrics: Zero-Deviation and Self-Consistency (Diag–CFM specific), and Ensemble Variance and FM Loss (general-purpose baselines). For each, we report Pearson correlation $\rho$ between uncertainty and ground-truth error, and percentage improvement over random selection.

**Results.** The Diag–CFM specific metrics consistently outperform general-purpose methods:

*Table 11.* Select-best performance across all datasets. For each target, $K=10$ candidates are generated and we select the one with lowest uncertainty. We report Pearson correlation between uncertainty and ground-truth error, and % improvement over random selection. Positive improvement means lower error than random. Zero-Deviation and Self-Consistency (Diag–CFM specific) consistently outperform general-purpose metrics across all datasets.

| | ZERO-DEV. | | SELF-CONS. | | ENS. VAR. | | FM LOSS | | ORACLE |
|---|---|---|---|---|---|---|---|---|---|
| DATASET | $\rho$ | IMPR. | $\rho$ | IMPR. | $\rho$ | IMPR. | $\rho$ | IMPR. | IMPR. |
| GAS TURBINE | **0.60** | **+23%** | 0.58 | +16% | 0.07 | −17% | 0.08 | −5% | +86% |
| UNIFOIL | **0.21** | **+31%** | 0.13 | +28% | 0.14 | +16% | −0.04 | −3% | +94% |
| DTLZ $P=12$ | **0.45** | +13% | 0.37 | **+15%** | 0.17 | −4% | 0.15 | −2% | +84% |
| DTLZ $P=24$ | 0.20 | **+7%** | 0.25 | +5% | **0.30** | +3% | 0.07 | −2% | +82% |
| DTLZ $P=50$ | 0.38 | **+16%** | **0.42** | +16% | 0.10 | −3% | 0.06 | −3% | +83% |
| DTLZ $P=100$ | 0.20 | +28% | **0.32** | **+31%** | 0.02 | −5% | 0.01 | +2% | +81% |

*Table 12.* Error-rejection performance: % reduction in mean error when rejecting the most uncertain 20% of samples. Positive values indicate improvement over no rejection. Zero-Deviation and Self-Consistency consistently enable effective abstention, while Ensemble Variance fails at high dimensions (negative on DTLZ $P=50$).

| DATASET | ZERO-DEV. | SELF-CONS. | ENS. VAR. | FM LOSS | ORACLE |
|---|---|---|---|---|---|
| GAS TURBINE | **+27%** | +24% | +8% | +5% | +49% |
| UNIFOIL | +14% | +15% | **+18%** | +5% | +57% |
| DTLZ $P=12$ | **+20%** | +16% | +3% | +6% | +41% |
| DTLZ $P=24$ | +8% | +9% | **+13%** | +7% | +40% |
| DTLZ $P=50$ | **+17%** | +17% | −1% | +4% | +37% |
| DTLZ $P=100$ | +10% | **+12%** | +1% | +3% | +34% |

- **Zero-Deviation** achieves the highest correlation on Gas Turbine ($\rho=0.60$), Unifoil ($\rho=0.21$), and DTLZ $P=12$ ($\rho=0.45$), translating to substantial improvements over random selection (+23%, +31%, and +13% respectively).

- **Self-Consistency** performs comparably, achieving the best improvement at DTLZ $P=100$ (+31%) and strong performance across all datasets.

- **General-purpose metrics** (Ensemble Variance, FM Loss) show weak or negative correlations, often performing *worse* than random selection.

The oracle improvement (81–94%) indicates significant room for perfect selection, yet the Diag–CFM specific metrics capture a meaningful fraction of this potential.

## H.2. Error-Rejection

We evaluate abstention: rejecting the most uncertain samples and measuring mean error on retained samples. For each of 1,000 target labels, we generate a single design and compute uncertainty using all four metrics. We compute error-rejection curves for rejection rates from 0% to 50%, comparing against random and oracle rejection.

**Results.** Table 12 shows error reduction at 20% rejection:

- **Zero-Deviation** achieves 8–27% error reduction, with the strongest performance on Gas Turbine (+27%) and DTLZ $P=12$ (+20%).

- **Self-Consistency** performs comparably, achieving the best results at DTLZ $P=100$ (+12%) where other metrics struggle.

- **Ensemble Variance** shows inconsistent behavior: reasonable on Unifoil (+18%) and DTLZ $P=24$ (+13%), but *fails* at high dimensions with negative error reduction on DTLZ $P=50$ (−1%).

- **FM Loss** provides minimal improvement (3–7%), often barely exceeding random rejection.

The oracle upper bound (34–57%) shows that perfect uncertainty estimation could substantially improve generation quality through selective abstention.

## H.3. Out-of-Distribution Detection

To evaluate whether our uncertainty metrics can identify when the model is asked to generate designs for physically unrealizable performance targets, we conduct out-of-distribution (OOD) detection experiments. For each dataset, we generate unfeasible performance labels by sampling from the complement of the training distribution, generate designs from these labels using Diag–CFM, and compute uncertainty measures for each generated design.

**OOD Point Generation.** We use a grid-based sampling strategy in the label space. First, each label dimension is normalized to $[0, 1]$ using the training set min/max. We then create a dense grid (30–40 points per dimension) spanning $[-0.1, 1.1]$ in normalized coordinates, allowing points slightly outside the observed range. For each grid point, we compute the Euclidean distance to the nearest training sample in this normalized space. OOD points are selected based on this *normalized distance*—a distance of 0.05 means the nearest training point is 5% of that dimension's range away. For the main OOD experiments we select points with normalized distances in $[0.02, 0.08]$. This creates a challenging detection task: OOD samples are geometrically close to the training boundary (within 2–8% of each dimension's range from the nearest training point) yet clearly outside the training support.

**Distance Sweep.** To calibrate this difficulty level, we also evaluate medium-distance OOD targets with nearest-neighbor distances in $[0.05, 0.15]$ and far-OOD targets in $[0.15, 1.0]$. Table 13 reports Zero-Deviation AUC for all three regimes. Across every dataset, AUC increases monotonically as targets move farther from the training support, confirming that the main hard setting is the most challenging distance-based OOD task.

*Table 13.* OOD distance sweep for Zero-Deviation. Entries report OOD detection AUC under the same Euler-30 setting used in the main OOD experiments. Easy, medium, and hard correspond to normalized nearest-neighbor distance ranges of 15–100%, 5–15%, and 2–8% of the label range, respectively. The main paper reports the hard near-boundary setting.

| DATASET | EASY | MEDIUM | HARD |
|---|---|---|---|
| GAS TURBINE | 0.993 | 0.903 | 0.773 |
| UNIFOIL | 0.994 | 0.976 | 0.957 |
| DTLZ $P=12$ | 1.000 | 0.890 | 0.823 |
| DTLZ $P=24$ | 0.969 | 0.838 | 0.739 |
| DTLZ $P=50$ | 0.978 | 0.851 | 0.790 |
| DTLZ $P=100$ | 0.980 | 0.861 | 0.731 |

**Uncertainty Metrics.** We compare four uncertainty measures: (1) **Zero-Deviation**, the squared norm of the label portion of the synthesis output at $t=1$, which should be near zero for in-distribution samples; (2) **Self-Consistency**, the reconstruction error when passing generated designs through an analysis pass; (3) **Ensemble Variance** across 5 independently trained Diag–CFM models; and (4) **FM Loss**, the flow matching loss at $t=0.5$.

### H.3.1. GAS TURBINE COMBUSTOR

For the gas turbine dataset, we evaluate 2,500 in-distribution samples from the validation set and 2,500 OOD samples selected using the calibrated distance strategy described above.

Figure 7 shows violin plots comparing uncertainty measure distributions for in-distribution versus OOD samples. Figure 8 presents ROC curves for OOD detection, where **Zero-Deviation achieves the best performance (AUC = 0.75)**, followed by Self-Consistency (AUC = 0.69), FM Loss (AUC = 0.68), and Ensemble Variance (AUC = 0.62).

### H.3.2. UNIFOIL

For the Unifoil dataset, we evaluate 2,500 in-distribution samples from the validation set and 2,500 OOD samples. For each label, we randomly sample physical conditioning parameters (angle of attack and Mach number) from the validation distribution.

Figure 9 shows violin plots comparing uncertainty distributions, revealing clear separation between in-distribution and OOD samples. Figure 10 presents ROC curves showing strong discrimination: **Zero-Deviation achieves the best performance (AUC = 0.96)**, followed by FM Loss (AUC = 0.89), Self-Consistency (AUC = 0.87), and Ensemble Variance (AUC = 0.85).

We also evaluate four Unifoil OOD tests beyond nearest-neighbor label-space distance. Three create physically infeasible

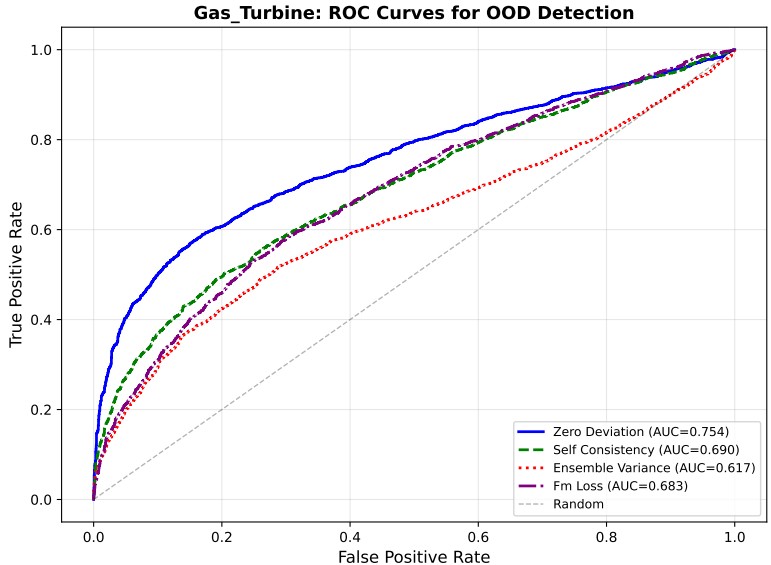

*Figure 7.* Violin plots comparing uncertainty measure distributions for in-distribution (green) versus out-of-distribution (red) samples on the Gas Turbine dataset. OOD samples are label-space points whose nearest training neighbor is 2–8% of each dimension's range away.

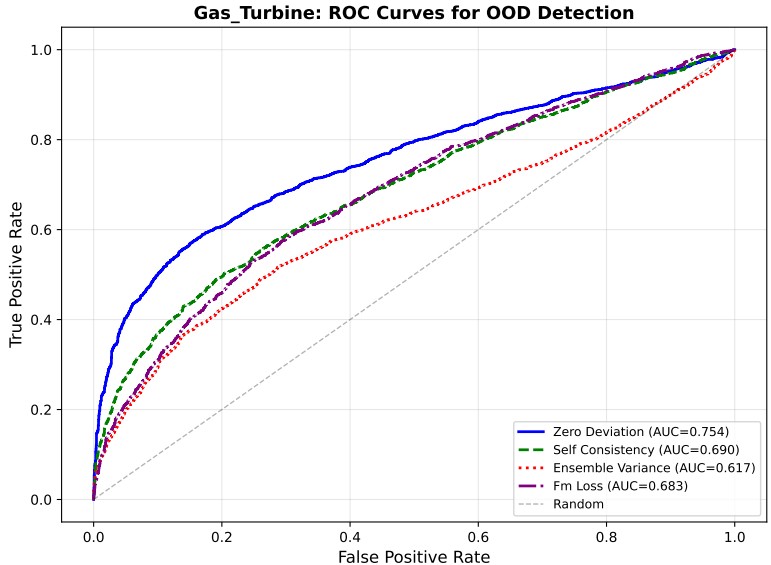

*Figure 8.* ROC curves for detecting out-of-distribution (unfeasible) performance labels on the Gas Turbine dataset. Zero-Deviation achieves AUC = 0.75, demonstrating the strongest OOD detection capability.

targets by perturbing real validation labels: negative drag flips $C_D$ below zero, extreme lift-to-drag requests L/D ratios in the range 200–500, and below-drag-polar lowers $C_D$ beneath an empirical drag-polar envelope while keeping each label dimension individually within the training range. The fourth test keeps target labels in-distribution but samples Mach number and angle of attack from bands extending 20% beyond the training range. Table 14 shows that Zero-Deviation remains strongly discriminative across all four settings, including the below-drag-polar case where infeasibility lies in the combination of lift and drag rather than a single out-of-range label.

*Table 14.* Unifoil OOD breadth tests. Entries report OOD detection AUC using 1000 in-distribution samples and 1000 OOD samples with Euler-30 evaluation. Higher is better.

| OOD TYPE | ZERO-DEV | SELF-CONS | ENS-VAR | FM LOSS |
|---|---|---|---|---|
| NEGATIVE DRAG | 0.996 | 0.952 | **0.999** | 0.992 |
| EXTREME L/D | **1.000** | 0.987 | 0.995 | **1.000** |
| BELOW DRAG POLAR | **0.988** | 0.897 | 0.826 | 0.763 |
| CONDITIONING SHIFT | **0.981** | 0.859 | 0.888 | 0.929 |

### H.3.3. DTLZ BENCHMARK

Table 15 shows OOD detection AUC scores for Diag–CFM across dimensions $P \in \{12, 24, 50, 100\}$. We evaluate 2,500 in-distribution and 2,500 OOD samples per configuration. **Zero-Deviation consistently achieves the best or near-best**

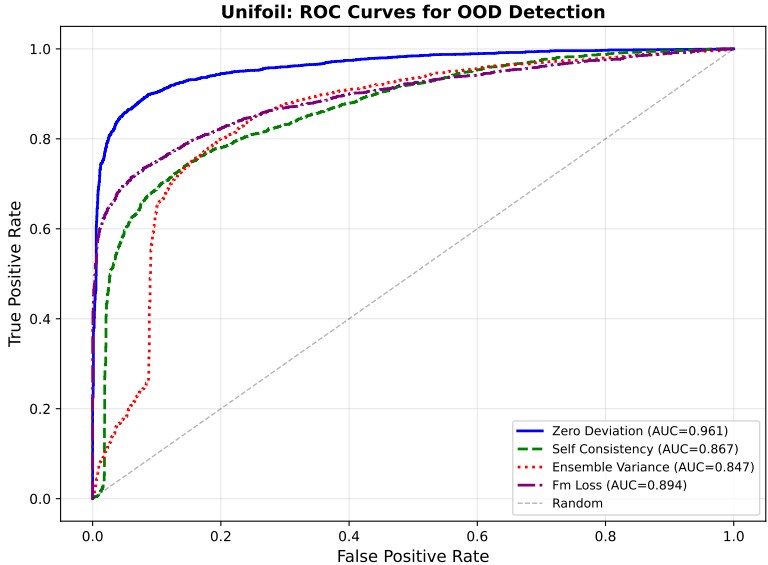

*Figure 9.* Violin plots comparing uncertainty measure distributions for in-distribution (green) versus out-of-distribution (red) samples on the Unifoil dataset.

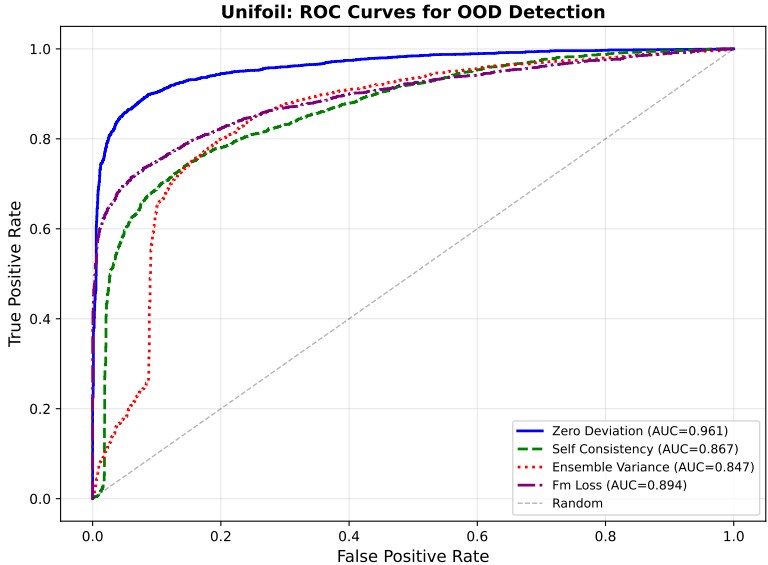

*Figure 10.* ROC curves for OOD detection on the Unifoil dataset. Zero-Deviation achieves AUC = 0.96, demonstrating excellent OOD detection despite sampling OOD points close to the training distribution boundary.

**OOD detection** across all dimensions, with AUC scores ranging from 0.73 to 0.82. Zero-Deviation and Self-Consistency maintain strong performance even at $P$=50, where Ensemble Variance begins to struggle.

*Table 15.* OOD detection AUC scores for DTLZ benchmark across design space dimensions.

| $P$ | Zero-Dev | Self-Cons | Ens-Var | FM Loss |
|-----|----------|-----------|---------|---------|
| 12  | **0.82** | 0.77      | 0.78    | 0.69    |
| 24  | **0.73** | 0.72      | 0.73    | 0.61    |
| 50  | **0.78** | 0.76      | 0.66    | 0.59    |
| 100 | **0.73** | 0.51      | 0.59    | 0.63    |

