# OpenReview forum: "Generative Inverse Design with Abstention via Diagonal Flow Matching"
_ICML.cc/2026/Conference — ICML 2026 regular_

### Official Review · Reviewer_PUvj · 2026-03-12

**Soundness:** 3
**Presentation:** 3
**Significance:** 4
**Originality:** 4
**Overall Recommendation:** 5
**Confidence:** 4

**Summary:**

Inverse design, as considered in this work, is concerned with finding optimal parameters for the performance of a more complex entity, e.g., optimizing air flow around a geometry. This is done by invertible conditional flow matching. The current work focuses on a problematic aspect of standard approaches in this context: the loss function (i) compares design coordinates with noise, (ii) it compares design coordinates with labels in a mixed way, making training very sensitive to the arbitrary ordering of coordinates. The introduced remedy is "zero-anchoring".

**Compliance With Llm Reviewing Policy:**

Affirmed.

**Key Questions For Authors:**

Fig. 3 is too tiny

The gas turbine dataset includes N_H as the number of fuel-injection holes as an integer. However, the paper treats all parameters as continuous. This seems to be a simplification that might be problematic when it comes to problems that are based on discrete variables (and where the discreteness matters). How serious is that?

Please address the above mentioned weakness concerning the dependency on the explicit Euler integrator scheme with a fixed 30-step, and in particular if it is worth moving away from this to more complex, e.g., adaptive schemes.

How does the performance scale if you are going to truly high dimensional problems?

**Strengths And Weaknesses:**

A clear strength of the work is the Diag-CFM formulation itself: the idea of augmenting the state space by L additional coordinats (such that labels are always paired with zero rather than with design coordinates). This is a simple, effective and non-obvious modification. Furthermore, the two architecture-related uncertainty metrics (zero-deviation and self-consistency) are also new, based on this zero-anchoring, allowing to formulate a permutation-equivariant learning problem (proposition 4.1).

The presented work is technically sound as it is based on two formal propositions with proofs; furthermore, experimental protocols are solid with regards to metrics and ensue fair comparisons. All information required for understanding the results were given. E.g., the authors are particularly careful about diversity metrics, demonstrating that raw diversity scores can be misleading which is useful for judging the results.

The authors demonstrate the applicability of their method based on various engineering inverse design problems. There, the experimental results showed a significantly reduced error. The work has a high significance as the developed method is applicable to various problems: whenever one needs to learn a bidirectional mapping between a high-dimensional parameter space and a low-dimensional performance or label space. Examples are: drug and molecular design, topology optimization as well as materials design.

The overall presentation is adequate both in terms of writing style and the artwork.

One of the weaknesses not mentioned by the authors is the dependency on the explicit Euler integrator scheme with a fixed 30-step discretization. Furthermore, all experiments are based on relatively simple MLP architectures is not clear how the scaling behavior to larger problems and more complex architectures is.

---

> ### Author Rebuttal · Authors · 2026-03-31
>
> We thank the reviewer for the positive assessment and for highlighting both the Diag-CFM formulation and the UQ metrics as key contributions. We address each point below, supported by new experiments, including a FashionMNIST Image Statistics benchmark (P=784, L=5), for which we use a U-Net CNN with self-attention.
>
> **Q1 — Figure 3.** We will increase the figure size in the revised paper.
>
> **Q2 — Discrete variables.** For N_H specifically, the continuous relaxation is standard practice — the forward function varies smoothly with N_H, and rounding to the nearest integer at deployment is straightforward. More broadly, this limitation is shared by all continuous generative models (CFM, INN, diffusion), not specific to Diag-CFM. For problems where discreteness fundamentally changes the design landscape (e.g., categorical material choices, combinatorial topology), the diagonal formulation could be adapted to discrete flow matching frameworks; this is an interesting direction for future work.
>
> **Q3 — Euler integrator dependency.** We evaluated all models across 3 solvers (Euler, RK4, Dopri5 adaptive) and 4 step counts (20, 30, 50, 100) — 9 configurations total (this analysis will be added to the appendix). The method ranking is identical across all configs. RK4 converges by 20 steps and matches Dopri5 (atol=rtol=1e-5) to 4+ significant figures. The following table shows gas turbine results; the same pattern holds across all datasets:
>
> | Method | Forward MSE range | Spread | Round-trip range | Spread |
> |--------|-------------------|--------|------------------|--------|
> | Diag-CFM | 1.55e-3 – 1.68e-3 | 8% | 1.32e-3 – 1.42e-3 | 8% |
> | CFM | 1.54e-2 – 2.05e-2 | 33% | 1.68e-2 – 2.49e-2 | 48% |
>
> We can see that Diag-CFM is much less sensitive to solver choice than CFM (8% vs 33–48%), consistent with its diagonal coupling creating straighter flow paths. We also verified that all four UQ metrics for OOD detection are essentially invariant to the solver (Zero-Deviation AUC spread: 0.3% across all 9 configs).
>
> **Q4 — Scalability and complex architectures.** We conducted two new benchmarks (both will be included in Section 5 of the paper).
>
> *FashionMNIST Image Statistics* (P=784, L=5) directly addresses both concerns. The task is inverse design on images: given 5 target nonlinear image statistics, generate a 28×28 garment image that achieves them. The forward map is exact and analytical — no surrogate approximation.
>
> | Label | Formula | Nonlinearity |
> |-------|---------|--------------|
> | Mean intensity | mean(x) | Linear |
> | Gradient energy | mean(|∇x|²) | Quadratic |
> | Variance of variances | Var(local 3×3 patch variances) | Quartic |
> | Pixel kurtosis | E[(x−μ)⁴]/σ⁴ | Quartic |
> | Log-contrast | log(mean_bright/mean_dark) | Logarithmic |
>
> The velocity network is a U-Net encoder-decoder (3 levels, channels 64/128/256) with skip connections, FiLM conditioning from label state and sinusoidal time embedding at every ResBlock, and self-attention at the 7×7 bottleneck (~4.3M params) — qualitatively different from the MLPs used in the paper. Results are mean ± std over 5 seeds:
>
> | Method | Forward MSE | Round-trip Error | FID |
> |--------|-------------|------------------|-----|
> | Diag-CFM | 0.001 ± 0.000 | 0.009 ± 0.002 | 13.4 ± 0.2 |
> | CFM | 0.024 ± 0.004 | 0.016 ± 0.005 | 19.4 ± 0.5 |
> | INN | 0.002 ± 0.000 | 0.027 ± 0.003 | 202.4 ± 1.0 |
>
> All labels are normalized to [0, 1]. Already at 50 epochs, images generated by Diag-CFM and CFM are visually almost indistinguishable from real dataset images, but Diag-CFM is the only method that successfully solves the full task: CFM fails in forward prediction (0.02 MSE on normalized labels), and INN does not produce acceptable images (FID 202.4 vs 13.4), as its coupling-layer architecture cannot leverage spatial inductive biases natural for image data. When training is extended to 200 epochs (one checkpoint per method), Diag-CFM's FID improves further, and the advantage over CFM grows to 60× better forward MSE and 1.9× better round-trip error, indicating the performance gap is structural.
>
> *Thin Film Optical Coating* (P=128, L=64) uses the Transfer Matrix Method, a highly nonlinear forward map (trigonometric phase + matrix chain multiplication). Results are mean ± std over 5 seeds:
>
> | Method | Forward MSE | Round-trip Error |
> |--------|-------------|------------------|
> | Diag-CFM | 0.011 ± 0.000 | 0.022 ± 0.000 |
> | CFM | 0.019 ± 0.001 | 0.029 ± 0.000 |
> | INN | 0.024 ± 0.000 | 0.031 ± 0.000 |
>
> Both benchmarks use exact analytical ground truth — no surrogate approximation. These results demonstrate that Diag-CFM's advantage extends to dimensions an order of magnitude beyond the paper's maximum (P=784 vs P=100) and to domain-specific architectures with spatial inductive biases.

---

> > ### Author Rebuttal · Reviewer_PUvj · 2026-04-03
> >
> > Thank you for the additional details and studies which I believe are an important aspect in the paper (in particular those in Q4) -- all my questions are properly resolved.

---

### Official Review · Reviewer_G91Y · 2026-03-13

**Soundness:** 3
**Presentation:** 2
**Significance:** 2
**Originality:** 3
**Overall Recommendation:** 4
**Confidence:** 3

**Summary:**

This paper studies data-driven inverse design where a high-dimensional design vector should be generated conditioned on lower-dimensional performance labels, while also enabling a reverse “analysis” direction that predicts y from x. The authors build a bidirectional invertible conditional flow matching approach and argue that a standard conditional flow matching (CFM) formulation can be unstable for inverse problems due to sensitivity to coordinate ordering (and scale). To address this, the paper proposes Diagonal Flow Matching (Diag–CFM), a “zero-anchoring” augmentation that aims to make the learning problem provably independent of coordinate permutations and empirically more stable. In addition, the paper introduces two architecture-specific uncertainty metrics, Zero-Deviation (computed from the anchored-to-zero components at the synthesis endpoint) and Self-Consistency (re-analyzing a synthesized sample after replacing approximate zeros with exact zeros), enabling practical deployment behaviors: selecting the best candidate among multiple generations, abstaining from unreliable predictions, and detecting out-of-distribution targets. The approach is evaluated on three inverse-design tasks (gas turbine combustor, airfoil aerodynamics, and DTLZ2 benchmark).

**Compliance With Llm Reviewing Policy:**

Affirmed.

**Final Justification:**

Most of my initial concerns are addressed by the authors during the rebuttal phase. I therefore raise my score accordingly.

**Key Questions For Authors:**

1. For gas turbine and Unifoil, round-trip error uses a trained surrogate forward model. How sensitive are the conclusions to surrogate error? Can the authors provide (i) stronger validation against a subset of true simulations, and/or (ii) robustness checks showing the ranking of methods is stable under surrogate uncertainty?
2. OOD targets are chosen by nearest-neighbor distances in a normalized label space (2–8% of each dimension’s range). How does performance change for (a) farther OOD, (b) OOD driven by physically infeasible constraints, or (c) shifts in conditioning variables (e.g., Mach/AoA in Unifoil)?
3. The method uses an explicit Euler integrator with 30 steps. How sensitive are results (round-trip error and especially Zero-Deviation) to solver choice and step count? Is Zero-Deviation partly measuring numerical integration error rather than purely epistemic uncertainty?
4. Have the authors compared against other conditional generative inverse-design approaches (e.g., diffusion-based conditional generation, or flow/diffusion with constraint-guidance) under comparable compute? If not, it would help to clarify the intended scope and why INN+CFM are the most relevant baselines here.

**Limitations:**

Yes.

**Strengths And Weaknesses:**

# Strengths:
- The paper clearly formalizes the inverse-design setup (many-to-one forward map, need for latent degrees of freedom) and motivates the need for a bidirectional model (generation + analysis) with uncertainty signaling.
- The proposed UQ metrics are concretely defined and leverage the model structure: Zero-Deviation is essentially “free” at generation time, and Self-Consistency is a meaningful internal check that avoids trivial inversion by snapping the near-zeros to exact zeros before analysis.
- Experiments include not only accuracy but also application-motivated behaviors: select-best, abstention/error-rejection, and OOD detection, with reported quantitative gains for these tasks in the main narrative.

# Weaknesses:
- Surrogate “ground truth” evaluation for gas turbine and Unifoil: round-trip error is computed via a pretrained surrogate (the paper describes this explicitly). This is understandable in expensive simulators, but it weakens the strength of the empirical claim unless surrogate fidelity is extensively validated and sensitivity is shown.
- OOD generation is defined via nearest-neighbor distance thresholds (2–8% of range per dimension). This is a reasonable “hard near-boundary” test, but it is also quite specific; it is unclear how performance translates to more realistic OOD shifts (e.g., far OOD, physically infeasible constraints, or shifts in conditional variables).

---

> ### Author Rebuttal · Authors · 2026-03-31
>
> We thank the reviewer for the constructive feedback. We address each question below, supported by new experiments, including two new benchmarks with exact analytical ground truth: Thin Film Optical Coating (P=128, L=64) and FashionMNIST (P=784, L=5).
>
> **Q1 — Surrogate sensitivity.** We provide four layers of evidence that surrogate error does not affect our conclusions:
>
> (1) Three benchmarks use exact analytical ground truth with no surrogate: DTLZ (in the paper), plus Thin Film (P=128, L=64, Transfer Matrix Method) and FashionMNIST (P=784, L=5, nonlinear image statistics). Diag-CFM achieves the best round-trip error across all five benchmarks. Thin Film tests a highly nonlinear forward map (trigonometric phase + matrix chain multiplication). FashionMNIST defines an inverse design task on images with exact analytical labels. Results are mean ± std over 5 seeds:
>
> | Benchmark (P, L) | Method | Forward MSE | Round-trip Error |
> |---|---|---|---|
> | Thin Film (128, 64) | Diag-CFM | 0.011 ± 0.000 | 0.022 ± 0.000 |
> | | CFM | 0.019 ± 0.001 | 0.029 ± 0.000 |
> | | INN | 0.024 ± 0.000 | 0.031 ± 0.000 |
> | FashionMNIST (784, 5) | Diag-CFM | 0.001 ± 0.000 | 0.009 ± 0.002 |
> | | CFM | 0.024 ± 0.004 | 0.016 ± 0.005 |
> | | INN | 0.002 ± 0.000 | 0.027 ± 0.003 |
>
> Will be added to Section 5.
>
> (2) Surrogate error is orders of magnitude smaller than inter-method gaps (gas turbine: surrogate MSE ~9e-4 vs smallest method gap ~1.5e-2; Unifoil: surrogate MSE 2.5e-5 vs gap ~2.1e-3).
>
> (3) [New experiment] We injected Gaussian noise into surrogate outputs at 1–20% of label std and re-evaluated: the ranking is preserved at all noise levels for both datasets (will be added to the appendix).
>
> (4) [New experiment] We trained 9 structurally different surrogates for Unifoil (3 architectures × 3 seeds). The ranking holds for all 9; inter-method gaps exceed inter-surrogate variation by 2.3–133× (sign test p < 0.002) (will be added to the appendix).
>
> **Q2 — OOD definition breadth.** We address all three sub-questions with new experiments:
>
> (a) Distance sweep across easy (15–100% of label range), medium (5–15%), and hard (2–8%) regimes. AUC scales monotonically with OOD distance. The paper reports the hardest setting (near-boundary), so our claims are conservative. Zero-Deviation is the best metric at all distances (e.g., gas turbine: 0.993/0.903/0.773 for easy/medium/hard).
>
> (b) Physics-based OOD on Unifoil with three categories of infeasible targets — negative drag, extreme lift-to-drag ratio (L/D 200–500), and below-drag-polar (every dimension is individually in-range, but the combination violates the drag polar). The last is hardest: no single dimension is out of range.
>
> (c) Conditioning-variable shift: extrapolated Mach/AoA (20% beyond training range) with in-distribution target labels. The model must detect that the flight conditions are unfamiliar, not the requested performance.
>
> | OOD type | Zero-Dev | Self-Cons | FM Loss | Ens-Var |
> |---|---|---|---|---|
> | Negative drag | 0.996 | 0.952 | 0.992 | 0.999 |
> | Extreme L/D | 1.000 | 0.987 | 1.000 | 0.995 |
> | Below drag polar | 0.988 | 0.897 | 0.763 | 0.826 |
> | Cond. shift | 0.981 | 0.859 | 0.929 | 0.888 |
>
> Zero-Deviation detects all four types with AUC > 0.98. Its advantage is largest on the hardest task (below-drag-polar), where it leads Self-Consistency by 9 points and FM Loss by 22 points. Will be added to the main body and appendix.
>
> **Q3 — Solver sensitivity & Zero-Deviation confound.** We evaluated all models across 3 solvers (Euler, RK4, Dopri5) and 4 step counts (20, 30, 50, 100) — 9 configurations total (appendix). Rankings are identical across all configs. Notably, Diag-CFM shows far lower solver sensitivity than CFM: across all 9 configs, Diag-CFM's forward and round-trip errors vary by only 8%, compared to 33–48% for CFM (see response to Reviewer PUvj). We also evaluated all four UQ metrics on OOD detection (gas turbine) across all 9 configs:
>
> | Metric | AUC range | Spread |
> |---|---|---|
> | Zero-Dev | 0.772–0.774 | 0.3% |
> | Self-Cons | 0.690–0.701 | 1.6% |
> | Ens-Var | 0.609–0.617 | 1.3% |
>
> This confirms that Zero-Deviation measures a structural property of the flow endpoint geometry, not integration error.
>
> **Q4 — Additional baselines.** The intended scope is fully amortized, bidirectional inverse design: a single model that provides both generation (labels → designs) and forward prediction (designs → labels), with no per-query optimization. This bidirectionality is what enables the UQ metrics central to our contribution: Zero-Deviation and Self-Consistency both require the forward pass. INN and CFM satisfy both properties, making them the natural baselines. Constraint-guidance methods (e.g., DFlow-SUR) reintroduce per-query optimization and are unidirectional, and conditional diffusion models (e.g., cDDPM) are amortized but unidirectional. INN and CFM are therefore the most directly comparable baselines for evaluating our contribution.

---

> > ### Author Rebuttal · Reviewer_G91Y · 2026-04-03
> >
> > I thank the authors for the detailed replies. Most of my concerns are addressed. I will raise my score.

---

### Official Review · Reviewer_CXuP · 2026-03-13

**Soundness:** 3
**Presentation:** 3
**Significance:** 3
**Originality:** 3
**Overall Recommendation:** 4
**Confidence:** 3

**Summary:**

This paper proposes Diagonal Conditional Flow Matching (Diag-CFM) for generative inverse design, arguing that standard conditional flow matching is not well aligned with inverse-design problems because its objective depends on arbitrary coordinate ordering and scaling between design and label spaces. The main idea is a simple zero-anchoring reformulation that pairs design dimensions with noise and label dimensions with zeros, which makes the method permutation-equivariant and substantially improves inverse-design performance. The paper also introduces two uncertainty measures, Zero-Deviation and Self-Consistency, to rank generated candidates, abstain on unreliable predictions, and detect out-of-distribution targets. Experiments on gas turbine combustor design, airfoil design, and DTLZ benchmarks show strong gains over standard CFM and INN baselines, while the proposed uncertainty measures also outperform several alternatives in candidate selection and rejection tasks.

**Compliance With Llm Reviewing Policy:**

Affirmed.

**Final Justification:**

The authors have addressed my main concerns, so I have decided to give this submission a positive score.

**Key Questions For Authors:**

please see weakness

**Limitations:**

yes

**Strengths And Weaknesses:**

Strengths:
- The paper identifies a meaningful failure mode of standard CFM for inverse design, and the proposed fix is simple, intuitive, and well motivated.
- The empirical improvements in inverse-design quality are strong and consistent across several datasets and dimensionalities.
- The uncertainty estimation methods are practical, lightweight, and useful for downstream tasks such as ranking, abstention, and OOD detection.

Weaknesses:
- Some experiments rely on surrogate forward models rather than true simulators, which weakens the claim of practical performance in real engineering settings.
- The method is especially strong for inverse design, but the results suggest it is not uniformly better for forward prediction, which limits its broader bidirectional advantage.

---

> ### Author Rebuttal · Authors · 2026-03-31
>
> We thank the reviewer for the positive assessment. We address both weaknesses below, supported by new experiments, including two new benchmarks with exact analytical ground truth: Thin Film Optical Coating (P=128, L=64) and FashionMNIST Image Statistics (P=784, L=5, CNN architecture).
>
> **W1 — Surrogate ground truth.** We provide four layers of evidence that surrogate error does not affect our conclusions:
>
> (1) Three of five benchmarks use exact analytical ground truth with no surrogate: DTLZ (in the paper), plus the two new benchmarks. The Thin Film benchmark uses the Transfer Matrix Method (TMM): given a stack of 128 alternating SiO₂/TiO₂ dielectric layers with thicknesses in [10, 300] nm, the TMM computes the reflectance spectrum at 64 wavelengths (400–800 nm) via trigonometric phase accumulation per layer, matrix chain multiplication across all layers, and modulus squared to obtain reflectance. This is an exact analytical computation — no numerical approximation or surrogate. The inverse task (finding layer thicknesses that produce a target reflectance spectrum) is challenging due to the highly nonlinear, many-to-one nature of the forward map. Thin film optical coating design is a canonical inverse problem in photonics, widely studied in both optimization and deep learning literature. All design parameters and labels are normalized to [0, 1]. Results are mean ± std over 5 seeds:
>
> | Method | Forward MSE | Round-trip Error |
> |--------|-------------|------------------|
> | Diag-CFM | 0.011 ± 0.000 | 0.022 ± 0.000 |
> | CFM | 0.019 ± 0.001 | 0.029 ± 0.000 |
> | INN | 0.024 ± 0.000 | 0.031 ± 0.000 |
>
> Diag-CFM achieves the best forward and round-trip performance on this benchmark.
>
> The FashionMNIST benchmark computes 5 exact nonlinear image statistics (gradient energy, kurtosis, log-contrast, etc.) from 28×28 images, using a U-Net CNN velocity network (~4.3M params). Diag-CFM also achieves the best performance on this benchmark (mean ± std over 5 seeds; see our response to Reviewer PUvj for full details):
>
> | Method | Forward MSE | Round-trip Error | FID |
> |--------|-------------|------------------|-----|
> | Diag-CFM | 0.001 ± 0.000 | 0.009 ± 0.002 | 13.4 ± 0.2 |
> | CFM | 0.024 ± 0.004 | 0.016 ± 0.005 | 19.4 ± 0.5 |
> | INN | 0.002 ± 0.000 | 0.027 ± 0.003 | 202.4 ± 1.0 |
>
> Diag-CFM achieves the best round-trip error across all five benchmarks. Full results will be added to Section 5 of the main body of the paper.
>
> (2) Surrogate error is orders of magnitude smaller than inter-method gaps (gas turbine: surrogate MSE ~9e-4 vs smallest method gap ~1.5e-2; Unifoil: surrogate MSE 2.5e-5 vs gap ~2.1e-3).
>
> (3) [New experiment] We injected Gaussian noise into surrogate outputs at 1–20% of label std and re-evaluated: the ranking is preserved at all noise levels for gas turbine and Unifoil. Even at 20% noise, Diag-CFM remains below the next-best method at 0% noise (this new experiment will be added to the appendix).
>
> (4) [New experiment] We trained 9 structurally different surrogates for Unifoil (3 architectures × 3 seeds, varying width 256–768, depth 4–8, and activation). The ranking holds for all 9; inter-method gaps exceed inter-surrogate variation by 2.3–133× (sign test p < 0.002) (will be added to the appendix).
>
> **W2 — Forward prediction at high dimensions.** We acknowledge that INN achieves lower forward MSE on DTLZ at P≥50. However, this is the only setting where this occurs, across all other benchmarks, Diag-CFM is the best forward predictor:
>
> | Dataset | P | Best forward | Best inverse |
> |---------|---|:------------:|:------------:|
> | Gas Turbine | 6 | Diag-CFM | Diag-CFM |
> | Unifoil | 20 | Diag-CFM | Diag-CFM |
> | DTLZ | 12 | Diag-CFM | Diag-CFM |
> | DTLZ | 24 | Diag-CFM | Diag-CFM |
> | DTLZ | 50 | INN | Diag-CFM |
> | DTLZ | 100 | INN | Diag-CFM |
> | Thin Film [new] | 128 | Diag-CFM | Diag-CFM |
> | FashionMNIST [new] | 784 | Diag-CFM | Diag-CFM |
>
> The two new benchmarks are particularly relevant: Thin Film (P=128, L=64) and FashionMNIST (P=784, L=5) are both higher-dimensional than the DTLZ settings where INN leads, yet Diag-CFM wins on both forward and inverse metrics. FashionMNIST has an even more extreme P/L ratio (784/5) than DTLZ-100 (100/3), yet Diag-CFM achieves the lowest forward MSE. This indicates that the DTLZ exception is benchmark-specific rather than a general high-dimensional limitation. (Our response to Reviewer PUvj contains further details on the FashionMNIST benchmark.)

---

> > ### Author Rebuttal · Reviewer_CXuP · 2026-04-01
> >
> > Thanks for the rebuttal. It have addressed my concerns, so I will keep my score to recommend the acceptance.

---

### Decision · Program_Chairs · 2026-04-30

**Decision:**

Accept (regular)

**Comment:**

This paper received a positive consensus after rebuttal, with reviewers finding the core idea technically sound and empirically strong. The main contribution—a zero-anchored diagonal conditional flow-matching formulation that removes sensitivity to arbitrary coordinate ordering in inverse design—was viewed as simple, well motivated, and effective. The rebuttal substantially strengthened the submission by adding exact-ground-truth benchmarks, robustness checks for surrogate evaluation, and solver-sensitivity analyses, which together address most of the reviewers’ original concerns.

One issue remains insufficiently discussed: the paper is not well positioned relative to the simulation-based inference literature. The task is closely related to conditional posterior or inverse inference, and there is relevant SBI work on flow-matching-based conditional generation; accordingly, the contribution is best understood as a specialized and practically useful refinement of modern conditional flow matching rather than a wholly separate direction. This gap does not negate the paper’s merits, but the final version should explicitly discuss its relationship to SBI and moderate the breadth of its novelty claims. Overall, I support accept.

References:
Dax and al. 2023: Flow Matching for Scalable Simulation-Based Inference